# Washing The Unwashable : On The (Im)possibility of Fairwashing Detection

**Ali Shahin Shamsabadi***
The Alan Turing Institute
Vector Institute

**Mohammad Yaghini***
University of Toronto
Vector Institute

**Natalie Dullerud***
University of Toronto
Vector Institute

**Sierra Wyllie**
University of Toronto
Vector Institute

**Ulrich Aïvodji**
ÉTS Montréal

**Aisha Alaagib**
University of Toronto
Vector Institute

**Sébastien Gambs**
Université du Québec à Montréal

**Nicolas Papernot**
University of Toronto
Vector Institute

## Abstract

The use of black-box models (*e.g.*, deep neural networks) in high-stakes decision-making systems, whose internal logic is complex, raises the need for providing explanations about their decisions. Model explanation techniques mitigate this problem by generating an interpretable and high-fidelity surrogate model (*e.g.*, a logistic regressor or decision tree) to explain the logic of black-box models. In this work, we investigate the issue of fairwashing, in which model explanation techniques are manipulated to rationalize decisions taken by an unfair black-box model using deceptive surrogate models. More precisely, we theoretically characterize and analyze fairwashing, proving that this phenomenon is difficult to avoid due to an irreducible factor—the unfairness of the black-box model. Based on the theory developed, we propose a novel technique, called FRAUD-Detect (FaiRness AUDit Detection), to detect fairwashed models by measuring a divergence over subpopulation-wise fidelity measures of the interpretable model. We empirically demonstrate that this divergence is significantly larger in purposefully fairwashed interpretable models than in honest ones. Furthermore, we show that our detector is robust to an informed adversary trying to bypass our detector. The code implementing FRAUD-Detect is available at https://github.com/cleverhans-lab/FRAUD-Detect.

## 1 Introduction

The wide applicability of machine learning models has recently increased their usage in high-stakes decision systems such as credit scoring [43], insurance risk [10] and predictive justice [30]. The consequences of erroneous decisions that are based on predictions of machine learning models (*e.g.*, people being wrongly denied parole [46]) have increased the demand—from both the public and government—to provide an explanation to humans about model decisions. For instance, this appears as an explanation requirement in the European General Data Protection Regulation [26]. However, various widely-used model architectures, such as deep neural networks, are considered black-boxes due to their complex and hidden internal logic, impeding the ability to explain their decisions in terms that are understandable by a human. To address this issue, black-box model

---

*Contributed equally.

36th Conference on Neural Information Processing Systems (NeurIPS 2022).

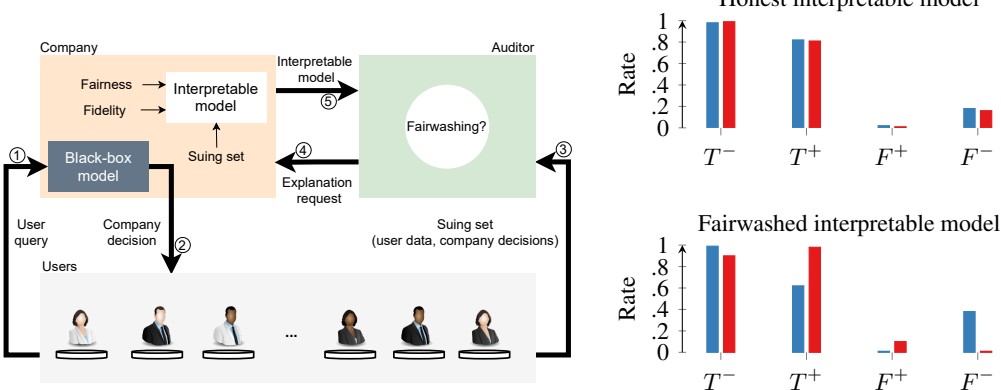

Figure 1: A dishonest company provides a high-fidelity but fairwashed interpretable model to rationalize decisions made by their unfair black-box model. FRAUD-Detect can be exploited by the auditor to distinguish between the fairwashed and honest interpretable model in a non-cooperative way with only query access to models. FRAUD-Detect detects fairwashing based on the agreement and disagreement of the interpretable model with the black-box model among subpopulations $0$ (🟦) and $1$ (🟥) quantified by the subpopulation-wise confusion matrix including the True Positive $(T^+)$, False Positive $(F^+)$, False Negative $(F^-)$ and True Negative $(T^-)$ rate of the interpretable model w.r.t. the black-box predictions. The top (bottom) plot shows that the confusion matrix of subpopulations are similar (different) in the honest (fairwashed) interpretable model. Fairwashing changes $T^+$ or $F^+$ in the opposite direction across subpopulations to equalize the probability of predicting positive label across subpopulations $0$ and $1$. For example, the fairwashed interpretable model decreases $T^+$ of subpopulation $0$ while increases $T^+$ of subpopulation $1$ with respect to their $T^+$s in the honest one.

explanation techniques [27, 6] aim to provide reasons for predictions of black-box models in human-understandable terms. One possible way to realize this is to approximate the black-box model using an inherently interpretable model (*e.g.*, logistic regression or decision tree) whose decisions can be easily explained by design [27]. For instance, in logistic regression, the coefficients of the model directly represent the importance of each input feature, while a decision tree is composed of a set of explainable if-then decision rules related to the input features.

In this black-box model explanation setting, the interpretable model is trained with the objective that its predictions agree with the predictions of the black-box model (*i.e.*, high fidelity [20]). However, black-box model explanations can be manipulated to provide deceiving explanations, even if these explanations display a high fidelity with respect to the original black-box model. An example of such manipulation occurs when the explanations provided cover up the unfairness of the underlying black-box model, resulting in *fairwashing* [2, 3]. As a concrete example, a dishonest bank [4] could use a black-box model that denies a loan to a customer in a discriminative manner—based on a sensitive attribute such as race or gender identity—while providing an interpretable surrogate model promoting the false impression that the decisions are fair and based solely on non-sensitive features. These manipulations are made possible in part by the ambiguity in current regulations [26] when it comes to describing what constitutes a valid explanation [45]. Fairwashing is clearly an ethical issue as individuals who have unfairly received unfavorable decisions are deprived of a chance to contest the decision [3].

Due to the growing importance of fairness in machine learning [7], we focus on a scenario in which a dishonest entity attempts to perform fairwashing [2, 3] when providing model explanations to an auditor (*e.g.*, an external dedicated party). For example, if a company's users expect that the decisions they received were a result of model bias, they may send their data and received decisions to an auditor. The auditor may make a legal demand [8] on the company to provide an interpretable model to explain these users' received decisions (see Figure 1). In this case, a dishonest company has an incentive to manipulate the black-box model explanation technique, to under-report and occlude the unfairness of the black-box model under scrutiny to evade the potential consequences of unfair decision-making. The dishonest entity could thus mislead the auditor by rationalizing its decisions

using the corresponding fairwashed interpretable model. In this paper, we are the first to propose a method for detecting such dishonest model explanations. This is a challenging problem. Fairwashing cannot be detected through computing the explanation violation of the explainable model, such as fidelity of explainable models, as we demonstrate that the explainable model cannot achieve perfect fidelity with respect to the black-box model. In addition to this, measuring the differences between the fairness of the black-box model and the explainable model cannot help to detect fairwashing as there are several design choices (including fairwashing) that could lead to different fairness, thus non-intentional fairwashing.

Our primary contributions include (1) an extensive theoretical analysis of fairwashing and the fairness limitations in explanation techniques and (2) a novel approach for detecting, and thus deterring, fairwashing. Our proposed method, FRAUD-Detect (FaiRness AUDit Detection), formulates the problem of detecting fairwashing as a non-cooperative test; both to overcome possible dishonest behaviors in entities that are actively being audited and for communication efficiency. FRAUD-Detect operates in a realistic scenario: FRAUD-Detect does not require access to the black-box model provided the users calling for the audit have provided their data and received decisions, which is a realistic setting as often the model is not provided by the entity due to intellectual property and trade-secret concerns [39, 37]. Thus, FRAUD-Detect only relies on the predictions of the interpretable and black-box models. Both our theoretical and empirical analysis quantify the per-subpopulation, per-label fidelity of the interpretable model with respect to the black-box model.

Figure 1 shows confusion matrix distributions for two sensitive subpopulations before and after fairwashing. These matrices are constructed by comparing the interpretable to the black-box model predictions. Fairwashing induces a divergence between subpopulation values to conceal the unfairness of the black-box model. FRAUD-Detect leverages the Kullback–Leibler (KL) divergence between subpopulation confusion matrices to distinguish between honest and fairwashed interpretable models. For a comprehensive empirical study, we examine logistic regressors and decision trees as interpretable models with respect to black-box Deep Neural Networks, AdaBoost [24], Gradient Boosted Decision Trees [17] and Random Forests [13] on three benchmark datasets: COMPAS [5], Adult Income [22] and Bank Marketing [34].

We evaluate the strength of our detector in the presence of an *informed adversary* representing an informed dishonest entity with knowledge of FRAUD-Detect that seeks to fairwash *and* bypass detection by constraining the divergence over subpopulation-wise confusion matrices. Our empirical experiments quantify the capacity of this informed adversary by computing the achievable fairness gap and fidelity under the additional detection constraint. We provide additional theoretical support for the challenge of jointly satisfying fairwashing and evasion in appendices.

In summary, our main contributions are as follows:

- We characterize fairwashing via a theoretical analysis on the difference in subpopulation gap between the black-box model and the interpretable model, *fairwashing violation* (Section 3).

- Based on our fairwashing theory, we establish requirements for a *sufficient* fairwashing detection method and prove that fairwashing is impossible to avoid completely due to an *irreducible* component in the violation term.

- We introduce the first method for fairwashing detection (Section 4.2). Informed by our theoretical results, we observe that fairwashing causes disparate effect on the subpopulation-wise fidelity distributions of the interpretable model with respect to the black-box predictions. We leverage this observation to propose a non-cooperative black-box access method, FRAUD-Detect, which utilizes KL divergence on per-subpopulation confusion matrices to distinguish between fairwashed and honest interpretable models.

- Our empirical results demonstrate that FRAUD-Detect successfully detects fairwashing based on the KL divergence between subpopulation-wise confusion matrices (Section 6). We show that the divergence over subpopulation confusion matrices can vary by over $0.6$ between an honest and a fairwashed interpretable models.

- We illustrate the robustness of FRAUD-Detect against an informed adversary (*i.e.*, a dishonest entity who attempts to fairwash while evading detection). Our empirical results show that evading our detector comes at the cost of a significant increase in subpopulation gap, negating fairwashing. Specifically, a dishonest entity that jointly attempts to fairwash while evading detection only achieves parity gaps greater than $10\%$.

## 2 Preliminaries and Problem Formulation

### 2.1 Related Work

**Fairness.** Many different formal definitions of algorithmic fairness have been proposed in the machine learning community [35]. One of the main challenges of algorithmic fairness is the lack of consensus on a universally applicable fairness definition. More precisely, it has been formally demonstrated that many such measures are incompatible with each other (*i.e.*, they cannot be achieved jointly in some situations) [31]. In addition, their differences are also rooted in their underlying philosophical and moral assumptions [28]. Thus, the choice of a particular fairness metric is usually context dependent. Nonetheless broadly speaking, there are two main families of fairness definitions: individual fairness and group fairness [23]. Individual fairness posits that "similar individuals should be treated similarly". In contrast, group fairness strives to equalize statistical properties of classification outcomes across subpopulations created by partitioning the population based on a sensitive attribute $A$ such as gender. In our work, we focus on a particular notion of group fairness, demographic parity, also called statistical group parity, which refers to observing equal probability of positive label prediction over subpopulations. One of the reasons we rely on demographic parity as the fairness metric is that true labels will not be available in a realistic scenario (especially in fairwashing scenario described in Section 2.3). In general, the existence of true labels conflicts with the assumption that when requesting decisions (or explanations), users (or auditors) do not have access to these true labels [42]. Furthermore, using the predictions of a black-box model as the true labels is paradoxical in the sense that the black-box model is assumed to be unfair. Demographic parity enforces independence between the class predicted by the model and inclusion in a particular subpopulation.

**Definition 1** (Demographic parity [15]). *A classifier $\hat{Y}$ satisfies demographic parity with respect to sensitive attribute $A$ if*

$$\Pr[\hat{Y} = 1 | A = a] = \Pr[\hat{Y} = 1 | A = b] \qquad \forall a, b \in A.$$

**Global black-box explanation methods** focus on explaining the whole logic of black-box models by training an inherently interpretable surrogate model. We refer the interested reader to Appendix A for the description of other explanation methods. In terms of abusing explanations via fairwashing, Fukuchi et al., similarly to us, try to detect fairwashing but with a different setting [25]. Slack et al. shows an alternate method of fairwashing that evades local model explanation techniques in a setting that is comparable to ours but they do not attempt to detect fairwashing [44]. We explore these additional fairwashing settings in Appendix B.

### 2.2 Desiderata for Global Black-box Explanations

We consider the setting in which an entity learns a complex black-box model $B(\cdot)$ on a training set $(X_{\text{Tr}}, Y_{\text{Tr}}, A)$, in which $A$ represents a sensitive attribute. We maintain the setup from recent fairwashing literature [3, 2] and consider binary classifiers mapping $M$ features into a binary label $B : \mathbb{R}^M \mapsto \{0, 1\}$. We also consider binary-valued sensitive attributes, $A \in \{0, 1\}$. Note that we do not assume that $A$ is necessarily used during the training of $B(\cdot)$. Given user queries, the entity uses the predictions of $B(\cdot)$ as part of a high-stakes decision system. Later, a group of these users may request an explicit explanation from the entity due to concerns over improper use of the sensitive attribute. Such a scenario frequently arises in high-impact domains, such as bank loans and credit scoring. Let $X_{\text{sg}}$ be a suing set comprising the unlabeled data of users demanding an explanation for their particular outcomes (see Figure 1). Consequently, an external auditing entity requires the company to provide explanations in terms that are understandable to humans about the predictions of the black-box. Here, we will assume that the explanation will be provided in the form of a global explanation. This means that a simple interpretable model $I(\cdot)$ will be trained on a dataset $X$ labeled by querying the black-box model $B(\cdot)$ trained on $X_{\text{Tr}}$, such that $I(\cdot)$ accurately reflects and explains the logic of $B(\cdot)$ in terms that are understandable to humans.

Formalizing further the auditing desiderata for black-box model explanations, the interpretable model $I(\cdot)$ must accurately mirror: 1) the output predictions of the black-box model (*i.e.*, fidelity) and 2) the fairness (according to a pre-defined metric) of the black-box model.

**Fidelity.** The interpretable model should satisfy the fidelity criterion for any $X \sim \mathfrak{D}$. Fidelity (defined in Appendix C) is difficult to perfectly achieve and challenging to measure for an auditor over all $X \sim \mathfrak{D}$. Thus, we additionally introduce the notion of empirical fidelity.

**Definition 2** (Empirical Fidelity). *The empirical fidelity on a dataset $X = \{x_i\}_{i=1}^N$ including $N$ data points $x_i$ is defined as the relative accuracy of $I(\cdot)$ with respect to $B(\cdot)$ on $X$:*

$$EmpiricalFidelity(I, B; X) = \frac{1}{N} \sum_{i=1}^N \mathbb{1}(I(x_i) = B(x_i)), \tag{1}$$

*where the indicator $\mathbb{1}(\cdot, \cdot)$ outputs 1 if the output label of $I(\cdot)$ and $B(\cdot)$ are the same and 0 otherwise.*

**Fairness.** As the audit of $B(\cdot)$ is expressly requested due to fairness concerns with respect to $A$, the interpretable model should reflect the fairness violations or adherences of the black-box model.

### 2.3 Fairwashing Definitions

To evade legal consequences of decision-making based on improper use of a sensitive attribute, a dishonest company may perform *fairwashing*. Fairwashing with respect to a pre-defined fairness metric permits the dishonest company to learn an interpretable model that hides the unfair behaviour of their black-box model. In practice, fairwashing could also occur due to the fact that current regulations [26] do not define what constitutes a *valid explanation*, thus leaving the possibility for the model provider to choose the interpretable model that meets their needs [45]. Fairwashing can be quantified with respect to a particular fairness measure.

**Definition 3** (Fairwashing). *Let $\Gamma_I$ and $\Gamma_B$ define the fairness gaps of the interpretable model and black-box model, respectively, with respect to a pre-defined fairness metric. For example, in terms of demographic parity (Definition 1):*

$$\Gamma_I := \Pr\left[\hat{Y}_I = 1 \mid A = 0\right] - \Pr\left[\hat{Y}_I = 1 \mid A = 1\right],$$
$$\Gamma_B := \Pr\left[\hat{Y}_B = 1 \mid A = 0\right] - \Pr\left[\hat{Y}_B = 1 \mid A = 1\right]. \tag{2}$$

*We define the* fairwashing violation $\gamma > 0$ *as the difference between $\Gamma_I$ and $\Gamma_B$:*

$$\gamma := \Gamma_B - \Gamma_I. \tag{3}$$

**Assumption 1.** *The fairwashing violation $\gamma$ is non-negative. Remark that the case of $\gamma < 0$ is the opposite of fairwashing since the interpretable model is displaying a bigger fairness gap than that of the black-box model, thus a model owner has no incentive to employ such an interpretable model.*

## 3 Fairness Gap of the Black-Box Breeds Fairwashing

We analyze whether it is possible to eliminate the risk of fairwashing altogether. We characterize the fairwashing violation $\gamma$ and show that completely eliminating fairwashing is impossible: an interpretable model explaining an unfair black-box model always has a non-zero fairwashing violation.

**Theorem 1.** *Assume $\hat{Y}_B$ are the black-box model $B(\cdot)$ predictions, $\hat{Y}_I$ are the predictions of a surrogate (interpretable) model $I(\cdot)$ trained on $B(\cdot)$ outputs and $A$ is a sensitive attribute: 1) If $B(\cdot)$ does not satisfy demographic parity, completely eliminating fairwashing is impossible (i.e., $\gamma > 0$); 2) A detector for fairwashing measuring false-positive rates and true-positive rates of $I(\cdot)$ with respect to $B(\cdot)$ is sufficient.*

We provide a proof sketch here (see Appendix D for full proof).

*Proof Sketch.* Define $T_a^+$, the true-positive rate of $I(\cdot)$ with respect to $B(\cdot)$ on $X \sim \mathfrak{D}_a$, in which $\mathfrak{D}_a$ denotes the distribution over data with attribute $a \in A$. Define $F_a^+$, the false-positive rate of $I(\cdot)$ with respect to $B(\cdot)$ on $X \sim \mathfrak{D}_a$. Denote the differences between $T_0^+$ and $T_1^+$, and $F_0^+$ and $F_1^+$ as $\tilde{\delta}, \delta'$, respectively. We can express $\Gamma_I$ in terms of $\tilde{\delta}, \delta', T_1^+, F_1^+, \Gamma_B$ and $Y_B|A$ by expanding terms and using Bayes' formula:

$$\Gamma_I = \Gamma_B \left(T_1^+ - F_1^+\right) + \left(\tilde{\delta} - \delta'\right) \Pr\left[\hat{Y}_B = 1 \mid A = 0\right] + \delta' \tag{4}$$

Now that we have derived $\Gamma_I$ in terms of our desired terms, we can eliminate $\Gamma_I$ from the equation for $\gamma$. The resulting formula for $\gamma$ in terms of $\tilde{\delta}, \delta', T_1^+, F_1^+, \Gamma_B$ and $Y_B|A$ allows us to demonstrate **correctness**, **irreducibility** and **sufficiency**.

$$\gamma = \Gamma_B(1 + F_1^+ - T_1^+) - \delta' \Pr\left[\hat{Y}_B = 0 \mid A = 0\right] - \tilde{\delta} \Pr\left[\hat{Y}_B = 1 \mid A = 0\right]. \tag{5}$$

**Correctness.** Note that if $\Gamma_B = 0$ (*i.e.*, black-box model is fair) then no-fairwashing is present, and no detector should mistakenly report fairwashing. We can verify this with Equation (15). Under Assumption 1, we know that $\gamma \geq 0$ and as below terms are, by definition non-negative, thus:

$$\gamma = -\left(\delta' \Pr\left[\hat{Y}_B = 0 \mid A = 0\right] + \tilde{\delta} \Pr\left[\hat{Y}_B = 1 \mid A = 0\right]\right) \geq 0 \implies \tilde{\delta} = \delta' = 0.$$

**Irreducibility.** If $\Gamma_B \neq 0$, even if $\tilde{\delta} = \delta' = 0$, *i.e.* when true-positive and false-positive rates are equal over subpopulations ($T_0^+ = T_1^+$ and $F_0^+ = F_1^+$), there exists an irreducible fairwashing violation:

$$\gamma = \Gamma_B(1 + F_1^+ - T_1^+) > 0, \tag{6}$$

since $1 + F_1^+ - T_1^+ > 0$ except in trivial cases (see Appendix D for the proof). We further point out that $1 + F_1^+ - T_1^+$ is not a function of disparity in the interpretable model as the value does not measure disparity between subpopulations. Rather, $1 + F_1^+ - T_1^+$ is a function of the difference in true-positive and false-positive rates ($T_1^+$ and $F_1^+$, respectively) within a single subpopulation. The sole measure of disparity in the gap above is black-box $\Gamma_B$, which is a given constant.

**Sufficiency.** The remaining factors of the gap are only function of $\tilde{\delta}$ and $\delta'$ as $\Pr\left[\hat{Y}_B = 1 \mid A = 0\right]$ and $\Pr\left[\hat{Y}_B = 1 \mid A = 0\right]$ are also given constants. Therefore, a detector that measures $\tilde{\delta}$ and $\delta'$ is sufficient to detect fairwashing. $\qquad\square$

## 4  Detecting Fairwashing

### 4.1  Fairwashing In Practice

Eliminating the fairness gap of the black-box model would be the ideal outcome as it would also remove the possibility for a dishonest model provider to perform fairwashing. However, in practice absolute fairness is often unrealistic and even regulatory bodies tolerate small violations of fairness laws [18]. In Section 3, we showed that fairness violations result in fairwashing violations unless the interpretable model has a higher fairness gap than the black-box model, which would not be desirable from the point of view of the model provider. Therefore, not all fairwashing violations may have been intended by the model owners. These *involuntary* violations are in fact side-effects of the imperfect fidelity, which is inherent to optimizing the empirical fidelity (Definition 2) of the interpretable model on a training set $X_{\text{tr}} \sim \mathfrak{D}$,

$$\max \quad \texttt{EmpiricalFidelity}(I, B; X_{\text{tr}}).$$

In contrast, deliberate fairwashing is the result of a similar optimization problem purposefully constrained to provide better fairness on a target suing set $X_{\text{sg}} \sim \mathfrak{D}$:

**Definition 4** (Fairwashing Optimization [2]). *Given a black-box model $B(\cdot)$ and a suing set $X_{sg}$, fairwashing optimization is defined as learning an interpretable model $I(\cdot)$ from $B(\cdot)$ on $X_{sg}$ such that the interpretable model has 1) high fidelity with respect to the black-box model and 2) is less unfair than this black-box model:*

$$\begin{aligned} \max \quad & \textit{EmpiricalFidelity}(I, B; X_{sg}) \\ \textit{subject to} \quad & \textit{FairnessGap}(I; X_{sg}) \leq \epsilon, \end{aligned} \tag{7}$$

*in which $\epsilon$ is an upper bound on the fairness gap of the interpretable model on the suing set $\texttt{FairnessGap}(I; X_{sg})$, and $\texttt{FairnessGap}(\cdot, \cdot)$ constitutes a measure of fairness gap according to some fairness metric – generically measured by the subpopulation gap in a fairness metric. Note that $\epsilon < \texttt{FairnessGap}(B; X_{sg})$ must be less than the fairness gap of the black-box model on the suing set $\texttt{FairnessGap}(B; X_{sg})$ to cause fairwashing violation $\gamma > 0$ (see Definition 3).*

## 4.2 Proposed method: FRAUD-Detect

As demonstrated in Section 3, a fairwashing detection method that relies on the difference over subpopulations in true-positive and false-positive rates of the interpretable model w.r.t the black-box model ($\tilde{\delta}$ and $\delta'$ respectively) is sufficient. Inspired by this theoretical analysis, we propose our fairwashing detector, FRAUD-Detect, which locates fairwashing violations. FRAUD-Detect aims to distinguish between honest and fairwashed interpretable models approximating the unfair black-box model $B(\cdot)$ as described in Section 2.2 under the additional following constraints: 1) without white-box access to the black-box and interpretable models as well as; 2) independently of the cooperation of the model provider being audited. These constraints are introduced to conform to realistic scenarios in which entities desire to retain the confidentiality of their black-box model due to intellectual property rights and because they considered it as a company's asset.

Similarly to definitions of true-positive $T_a^+$ and false-positive $F_a^+$ rates of $I(\cdot)$ w.r.t $B(\cdot)$ on data with attribute $a \in A = \{0, 1\}$, we define the true-negative rate, $T_a^-$, and false-negative rate, $F_a^-$ as:

$$T_a^- = \Pr[\hat{Y}_I = 0 \mid \hat{Y}_B = 0, A = a], \quad F_a^- = \Pr[\hat{Y}_I = 0 \mid \hat{Y}_B = 1, A = a] \tag{8}$$

for $a \in A = \{0, 1\}$. Note that $T_a^- = 1 - F_a^+$ and $F_a^- = 1 - T_a^+$.

FRAUD-Detect computes the $T_a^+, F_a^+, T_a^-, F_a^-$ for each subpopulation $a \in A = \{0, 1\}$ using the predictions of the interpretable model and black-box model on the suing set $X_{\text{sg}}$. Let

$$C_0 = [T_0^+, F_0^+, T_0^-, F_0^-], \quad C_1 = [T_1^+, F_1^+, T_1^-, F_1^-], \tag{9}$$

be the (flattened) confusion matrices of the interpretable model w.r.t the black-box model on $X_{\text{sg}}$ with attribute 0 and 1, respectively. Then, FRAUD-Detect computes the divergence between $C_0$ and $C_1$ using Kullback–Leibler (KL) as:

$$\mathcal{C}_{\text{KL}} = \text{KL}(C_0, C_1). \tag{10}$$

Recall that we demonstrated in Section 3 that the divergence in $T_0^+$ and $T_1^+$ ($\tilde{\delta}$) and $F_0^+$ and $F_1^+$ ($\delta'$) were sufficient for a detection method. We can complement[2] our detection method with additional divergences in $T_a^-$ and $F_a^-$ and measure the dissimilarity via KL divergence. This choice is natural as the KL divergence is commonly used to quantify the divergence over probability distributions— $C_a$ functions as a simplified representation of the probability distribution $Y_I|Y_B, A = a$. For an honest interpretable model, $\mathcal{C}_{\text{KL}}$ is generally relatively low ($\tilde{\delta}, \delta' \approx 0$) as the only divergence arises from general error in fidelity optimization and the fairness gap of the black-box model (as shown in Section 3). In other words, the honest interpretable model approximates the black-box model equivalently across subpopulations. However, for a fairwashed interpretable model, $\mathcal{C}_{\text{KL}}$ grows significantly, as the interpretable model is explicitly manipulated to optimize the fairness across subpopulations and improve over the black-box model. We quantify the distinction between honest and fairwashed interpretable models via a threshold $\Delta > 0$ on $\mathcal{C}_{\text{KL}}$ such that if $\mathcal{C}_{\text{KL}} > \Delta$, the interpretable model is considered fairwashed. Note here that due to the irreducibility result stated in Section 3, $\Delta$ must be chosen such that $\Delta$ is tightly greater than the irreducible term in order to properly distinguish between accidental fairwashing that occurs as a result of the irreducibility and malicious fairwashing. This procedure is detailed in Algorithm 1.

# 5 Evading FRAUD-Detect

In this section, we investigate on whether a dishonest entity could evade FRAUD-Detect while performing fairwashing. We assume that this dishonest entity, which we call the *adversary*, is informed about FRAUD-Detect. Namely, the adversary is aware of the fairwashing detection method and desires to fairwash while evading detection by the auditor. To achieve this goal, the informed adversary has the objective of finding a fairwashed interpretable model that satisfies an additional constraint on the Kullback–Leibler (KL) divergence of the confusion matrices among subpopulations, $\mathcal{C}_{\text{KL}}$ in the previous section. To probe the robustness of our detector to an informed adversary empirically, we explore the range of fairness gap given a fixed value of fidelity and a fixed value of $\mathcal{C}_{\text{KL}}$ via solving the informed adversary optimization problem.

---

[2]We performed experiments in two settings: 1) considering all related true-positive, false-positive, true-negative and false-negative rates; 2) considering only independent true-positive and false-positive rates.

**Algorithm 1** FRAUD-Detect. $\{\texttt{predicate}\} = \{(x, a, \hat{y}_B, \hat{y}_I) \in (X_{\text{sg}}, A, \hat{Y}_B, \hat{Y}_I) : \texttt{predicate}\}$

**Input:** Query access to the interpretable model $I(\cdot)$, suing dataset $X_{\text{sg}}$, Black-box model predictions on the suing set $B(X_{\text{sg}})$, sensitive attribute $a \in A$ and a threshold $\Delta > 0$.

**Output:** $\begin{cases} T & \text{fairwashing is detected} \\ F & \text{fairwashing is not detected} \end{cases}$

1: $\hat{Y}_I \leftarrow I(X_{\text{sg}})$   ▷ Query interpretable model
2: **for** $i \leftarrow 0, 1$ **do**
3:     $T_i^+ \leftarrow \frac{|\{\hat{y}_B=1, \hat{y}_I=1, a=i\}|}{|\{\hat{y}_B=1, a=i\}|}$   ▷ TPR
4:     $F_i^+ \leftarrow \frac{|\{\hat{y}_B=0, \hat{y}_I=1, a=i\}|}{|\{\hat{y}_B=0, a=i\}|}$   ▷ FPR
5:     $F_i^- \leftarrow \frac{|\{\hat{y}_B=1, \hat{y}_I=0, a=i\}|}{|\{\hat{y}_B=1, a=i\}|}$   ▷ FNR
6:     $T_i^- \leftarrow \frac{|\{\hat{y}_B=0, \hat{y}_I=0, a=i\}|}{|\{\hat{y}_B=0, a=i\}|}$   ▷ TNR
7: $C_0 \leftarrow [T_0^+, F_0^+, T_0^-, F_0^-]$
8: $C_1 \leftarrow [T_1^+, F_1^+, T_1^-, F_1^-]$
9: $\mathcal{C}_{KL} \leftarrow \text{KL}(C_0, C_1)$   ▷ Kullback–Leibler
10: **if** $\mathcal{C}_{KL} > \Delta$ **then**
11:     **return** T
12: **else**
13:     **return** F

---

**Definition 5** (Informed Adversary Optimization). *Given a black-box model $B(\cdot)$, a suing set $X_{sg}$, a sensitive attribute $A$, a loss threshold $v$ and a fairwashing detection threshold $\Delta$, an informed adversary aims to learn an interpretable $I(\cdot)$ such that*

$$\begin{aligned} \textit{minimize} \quad & \texttt{FairnessGap}(I; X_{sg}) \\ \textit{subject to} \quad & L(I; X_{sg}) \leq v \quad \textit{and} \quad \mathcal{C}_{KL} \leq \Delta, \end{aligned} \tag{11}$$

*where $\mathcal{C}_{KL}$ is the KL divergence between the subpopulation confusion matrices of $X_{sg}$.*

The differences between the fairwashing optimization in Definition 4, and the informed adversary optimization of Definition 5 are two fold. First, the order of the maximand and the constraint is reversed. Both formulations are valid since each indicates priorities of the adversary, namely, whether to put a hard limit on the fidelity loss or on the fairness gap of the interpretable model. Second, in Definition 5 we replaced the non-differentiable $\texttt{EmpiricalFidelity}(I, B; X_{\text{tr}})$ with a differentiable measure of loss $L(I; X_{sg})$. In our empirical evaluations, we use the logistic regression loss.

To solve the above optimisation problem, we consider the Rashomon set of high-fidelity interpretable models and compute the range of the fairness gap of interpretable models. In short, given a classification task, the Rashomon Set [41] is defined as the set of almost-equally-accurate models (interpretable models in our case). More precisely, given a model class $\mathcal{F}$, a loss function $L_{\mathcal{D}}(\cdot)$ over a dataset $\mathcal{D}$ of interest, a reference model $f^*$ (*e.g.*, optimal model) and a performance threshold $\tau \in [0, 1]$, the Rashomon set $R_s(\mathcal{F}, f^*, \tau) = \{f \in \mathcal{F} \mid L_{\mathcal{D}}(f) \leq L_{\mathcal{D}}(f^*) + \tau\}$.

We extend the Fairness in the Rashomon Set (FaiRS) algorithm [19] to compute the range of the fairness gap of interpretable models that can be generated over the set of high-fidelity models satisfying a constraint on the KL divergence. FaiRS exploits the so-called Rashomon Effect [14], which is an empirical phenomenon resulting in multiple models displaying the same performance overall (*e.g.*, w.r.t their global accuracy) but have significant differences in terms of their individual predictions [19]. In our setting, the Rashomon effect implies that several interpretable models can achieve the same high fidelity w.r.t the black-box model while displaying different values of fairness gap. Thus, computing the range of the fairness gap of high-fidelity interpretable models that satisfy a constraint on $\mathcal{C}_{KL}$ quantifies the robustness of FRAUD-Detect to the informed adversary.

To the best of our knowledge, the FaiRS algorithm of [19] is the only work proposing a practical solution to efficiently characterize the range of the fairness gap over the Rashomon set. Alternative approaches would require measuring the range of the fairness gap over approximations of the Rashomon set obtained by either generating models using several hyperparameter values (*e.g.*, seeds [33, 21]) or brute-forcing over a particular class of model (*e.g.*, depth seven decision trees [41]). This is not practical due to the high computational cost and may lead to sub-optimal results.

## 6 Empirical validation

FRAUD-Detect aims to detect fairwashing by distinguishing between a fairwashed interpretable model and honest interpretable one during an audit. Additionally, FRAUD-Detect seeks to be robust

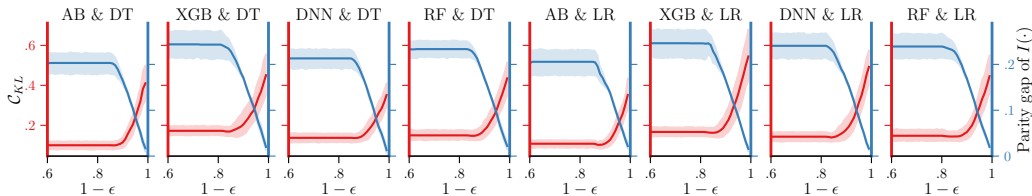

Figure 2: KL divergence between the confusion matrices of subpopulations, $\mathcal{C}_{KL}$, and parity gap as a function of $\epsilon$ fairwashing in interpretable models (Logistic Regression (LR) and Decision Tree (DT)) explaining black-box models (AB, Deep Neural Network (DNN), Random Forest (RF) and Gradient Boosted Decision Trees (XgBoost)) using COMPAS. Values before $1 - \epsilon = .6$ are constant and not shown. See Appendix F for the results of other two datasets.

even against an informed adversary, *i.e.*, demonstrate the ability to detect fairwashing even when a dishonest entity is aware of FRAUD-Detect and attempts evasion. Therefore, we empirically validate the performance of FRAUD-Detect in: 1) capturing a relationship between $\mathcal{C}_{KL}$ and demographic parity gap of the interpretable model $I(\cdot)$: $\mathcal{C}_{KL}$ increases as the demographic parity gap of the interpretable model decreases (*i.e.* as fairwashing becomes more severe); 2) restricting the fairwashing capabilities of the informed adversary: incorporation of $\mathcal{C}_{KL}$ constraint into fairwashing problem prevents favorable demographic parity gaps for the interpretable model, precluding fairwashing.

We assess the performance of FRAUD-Detect using a diverse set of black-box architectures, interpretable models and datasets. More precisely, we consider four architectures of black-box models: Deep Neural Networks (DNN), AdaBoost (AB) [24], Gradient Boosted Decision Trees (XGBs) [17] and Random Forests (RFs) [13]. We evaluate the approach on three real-world datasets corresponding to critical decision systems: Adult Income [22], Bank Marketing [34] and COMPAS [5]. We rely on Logistic Regression (LR) and Decision Trees (DTs) as our interpretable models due to their high performance and ease of explainability. We refer to Appendix E for details on these datasets, black-box models, interpretable models and implementation of FRAUD-Detect.

**FRAUD-Detect successfully detects fairwashing.** Figure 2 illustrates both the demographic parity gap and $\mathcal{C}_{KL}$ of interpretable models as a function of $\epsilon$. The bigger the value of $\epsilon$, the stronger the fairwashing (see Definition 4). Any graph within the figures may be read from left to right as the degree of fairwashing increases. As fairwashing becomes more severe, the demographic parity gap decreases while $\mathcal{C}_{KL}$ increases such that we observe a relationship between the demographic parity gap of interpretable models and the associated $\mathcal{C}_{KL}$ values. For example, decreasing the demographic parity gap of Decision Trees trained on a COMPAS dataset labelled by a RF black-box model by half, increases $\mathcal{C}_{KL}$ by a factor of 4. The amount of change in $\mathcal{C}_{KL}$ and in the demographic parity gap of fairwashed interpretable models differ across the dataset and black-box models due to the different original fairness of the black-box models. As $\mathcal{C}_{KL}$ is sensitive to changes in fairwashing, FRAUD-Detect can be used to detect fairwashing by not permitting $\mathcal{C}_{KL}$ to pass a certain threshold value. The $\mathcal{C}_{KL}$ thresholds corresponding to 5% and 50% fairwashing are reported in Appendix G. Per-seed results are provided in Appendix H.

**FRAUD-Detect is robust to the informed adversary.** Figure 3 shows the range of the demographic parity gap of high-fidelity fairwashed interpretable models subjected to a constraint on the $\mathcal{C}_{KL}$ by solving the optimization problem of Equation (11). We use four types of black-box models (AB, XGB, DNN and RF). However, for simplicity, we performed the experiment using logistic regression as the interpretable model. The fairwashing detection thresholds were chosen from a wide range, *i.e.* $\Delta \in \{0.01, 0.03, 0.05, 0.07, 0.10, 0.20\}$ (the constraint on $\mathcal{C}_{KL}$ in Equation (11)). A direct implication for FRAUD-Detect is that using low fairwashing detection thresholds makes fairwashing difficult. Consistently over all these results, imposing a constraint on the $\mathcal{C}_{KL}$ significantly narrows the range of demographic parity gap of fairwashed interpretable models. For instance, on COMPAS with $\Delta = 0.2$, for an AB model that has demographic parity gap of $0.214$, fairwashing can produce a completely fair (*i.e.*, demographic parity gap of $0.0$) high-fidelity interpretable model while remaining undetected (*i.e.*, $\mathcal{C}_{KL} < 0.2$). However, when $\mathcal{C}_{KL}$ decreases to $0.1$, the fairest high-fidelity interpretable model obtained by the adversary exhibits a demographic parity gap of $0.107$.

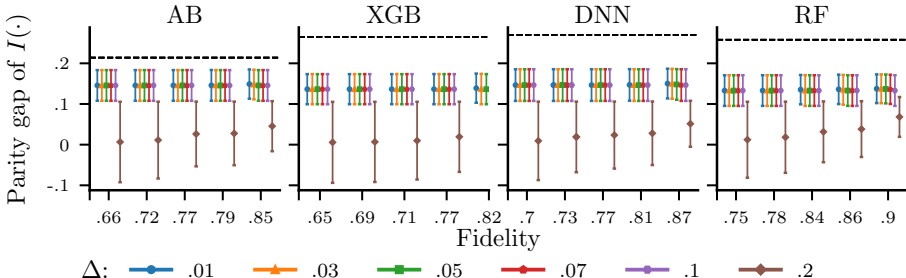

Figure 3: Range of the demographic parity gap of fairwashed logistic regression interpretable models subjected to a constraint ($\Delta$) on the $\mathcal{C}_{\text{KL}}$ ($\Delta \in \{0.01, 0.03, 0.05, 0.07, 0.10, 0.20\}$), explaining AB, Deep Neural Network (DNN), RF (RF) and Gradient Boosted Decision Trees (XgBoost) black-box models trained on using COMPAS. Horizontal lines denote the parity gap of the black-box models. See Appendix F for the results of other two datasets.

## 7 Conclusion and Future Work

Incorporating fairness and explainability in decision systems can help foster trust [29] in them by characterizing failure modes (*e.g.*, unfairness) and providing assurances (*e.g.*, fairness constraints) [16]. Auditing provides an oversight component and helps avoid *first-order failures* such as unfairness. In this sense, fairwashing can be seen as failure of auditing—a *second-order failure* that reduces trust in both fairness and explainability. Second-order failures help characterize the limits of techniques designed to reduce the risks of first-order failures. Our work is the first to delineate the theoretical limits of what is possible in auditing fairness and in containing the risk of fairwashing. In practice, we demonstrated the possibility of performing fairwashing through joint optimization of fidelity and fairness constraints when providing explanations. This could thwart auditing processes that are solely based on an analysis of the delivered interpretable model. We address this issue by proposing an additional auditing protocol that queries the interpretable model. Finally, through both a theoretical framework and an experimental evaluation, we demonstrate that our fairwashing detector cannot be evaded by an attacker, informed of our detector, without significantly degrading the fidelity of model explanations. Future directions include the extension of the detector to other fairness measures (*e.g.*, measures incorporating ground-truth information) as well as extending the analysis to multi-class setups. However, these would both require additional assumptions on the data manifold (to account for class correlations) to establish similar results as in Section 3.

## Acknowledgments

This work was supported by CIFAR (through a Canada CIFAR AI Chair and a Catalyst grant), by NSERC (under the Discovery Program, and COHESA strategic research network), by the Ontario Early Researcher Award, and by gifts from Intel and Meta. We also thank the Vector Institute's sponsors. Ali Shahin Shamsabadi was also partially supported by The Alan Turing Institute. Sierra Wyllie was also partially supported by Engineering Science at University of Toronto through ESROP grant. Ulrich Aïvodji was supported by the NSERC Discovery Grants program (2022-04006). Sébastien Gambs is supported by the Canada Research Chair program, a Discovery Grant from NSERC as well as the NSERC-RDC DEEL project.

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
