## Broader Impact Statement

FRAUD-Detect may be used to provide evaluations of the fairness of model explanations, resulting in increased trust in explanations, public expectation of model fairness, and model owner accountability. Since FRAUD-Detect is non-cooperative and resists evasion even against an informed adversary, it empowers auditors, protects the intellectual property and ownership of the model owners, and may resist gamification into a new arms-race system between fairwashing and auditor detection methods. By empowering auditors, FRAUD-Detect also enforces democratic ideals of equality by providing an avenue for users to receive fairness-vetted model explanations for high-stakes decisions, further increasing public trust in AI and model auditing.

The potential risks of this work revolve around the fairness definition and datasets used in this paper. Demographic parity, while befitting our setting (as discussed in Section 2.1) is not always an appropriate fairness definition, depending on the model's domain and application. We recognise that continual focus on a concept reinforces it and it's ideology's normalcy, and that our focus on demographic parity also reinforces the political and philosophical ideologies underlying it and deprives alternate definitions of due attention. Likewise, as noted in Appendix E, our evaluation of FRAUD-Detect on the current popular fairness benchmark datasets normalizes these applications of AI. We draw specific attention to the use of the COMPAS recidivism prediction dataset [5], and note that we do not necessarily condone the use of AI models for this application.

To mitigate these societal risks, we advocate for similar auditing measures for second-order failures with respect to other fairness definitions (see Section 7). We also acknowledge the importance of investigating the impact of FRAUD-Detect in real-world scenarios, especially where the group of users suing for model explanations represents minority or sensitive demographics.


# A   Explanation methods

Methods to facilitate the understanding of machine learning models are usually classified into two main categories: transparent-box design and black-box model explanation [27]. Transparent-box design concerns methods that produce inherently interpretable models such as logistic regressors and decision trees [12, 32]. Black-box model explanation corresponds to approaches designed to explain how black-box models produce their outcomes. For instance, global explanations techniques (*e.g.*, TREPAN [20] and DECTEXT [11]) focus on explaining the whole logic of black-box models by training an inherently interpretable surrogate model. On the other hand, local explanations aim at explaining black-box models decisions on single data points (*e.g.*, LIME [38], and GRADCAM [40]). For simplicity, in this paper, we focus on fairwashing performed with global explanation models.

# B   Related Work

We highlight some of the differences in settings, assumptions and solutions between our work and [44, 25]:

**Setting** We rely on explainer models while they use a published data subset for fairness auditing. In particular, they reveal some subset of the training data and their predictions. We follow the setting considered in the literature introducing fairwashing [2, 3]. Slack et al. explanations require input perturbations while ours require the model owner to provide an interpretable model.

**Assumptions** Slack et al. assume that fairness auditing is performed via model-agnostic local explanations (e.g., LIME and SHAP) [44]. Both [44] and [25] assume an ideal detector with knowledge of the underlying training distribution of the model; and query access to the black-box model. However, we only assume access to black-box model predictions on the suing set. We stress that this dataset is available before any formal audit takes place (in the form of asking for model explanations). This does *not* constitute query access to the black-box model; in fact it is only dataset access, which here is the predictions on the suing set.

**Solutions** [44, 25] attempt to detect fairwashing using a Kolmogorov-Smirnov (KS) test over the underlying and company-provided subset distribution to determine if the data subset was honestly provided. More precisely, they show that detection is difficult when fairwashing is conducted by minimizing the Wasserstein distance between the distributions while subject to fairness; this optimization also minimizes the upper-bound of the advantage (i.e., distinguishability) of the KS test. This differs from our setting in which an informed malicious company must optimize the fairness subject to both fidelity (loss) and the detection threshold. Explanation methods like LIME and SHAP perturb inputs to gauge feature relevance, inadvertently querying with synthetic data that may be detected with out-of-distribution detection. Queries determined to originate from explainers are fed to a fair model whereas in-distribution points are given to the biased model. In contrast, our detection method is non-cooperative (we require no additional information from the black-box model as the auditor already has the predictions on the suing set) and therefore does not rely on input perturbations.

# C   Fidelity

**Definition 6** (Fidelity [20]). *The fidelity of an interpretable model $I(\cdot)$ with respect to a black-box model $B(\cdot)$ is defined as the relative accuracy of $I(\cdot)$ with respect to $B(\cdot)$:*

$$\texttt{Fidelity}(I, B) = \Pr_{X \sim \mathfrak{D}}[\hat{Y}_B = \hat{Y}_I \mid X], \tag{12}$$

*where $\mathfrak{D}$ is a data distribution, $\hat{Y}_I$ is the prediction of the interpretable model and $\hat{Y}_B$ is the prediction of black-box model. $\hat{Y}_I$ and $\hat{Y}_B$ are assumed to be both binary-valued random variables.*

# D   Additional Proofs

Below we iterate the full proof for Theorem 1, the main theorem in the paper.

*Proof.* Define $T_a^+$, the true-positive rate of $I(\cdot)$ with respect to $B(\cdot)$ on $X \sim \mathfrak{D}_a$, in which $\mathfrak{D}_a$ denotes the distribution over data with attribute $a \in A$. Define $F_a^+$, the false-positive rate of $I(\cdot)$ with respect to $B(\cdot)$ on $X \sim \mathfrak{D}_a$:

$$T_a^+ = \Pr[\hat{Y}_I = 1 \mid \hat{Y}_B = 1, A = a], \quad F_a^+ = \Pr[\hat{Y}_I = 1 \mid \hat{Y}_B = 0, A = a] \quad \text{For} \quad a \in \{0, 1\}. \tag{13}$$

Note that by definition, $T_a^+, F_a^+ \in [0, 1]$. We denote $\tilde{\delta}, \delta' > 0$ as the difference in value between $T_0^+$ and $T_1^+$, and $F_0^+$ and $F_1^+$, respectively. Without loss of generality, let $T_1^+ = T_0^+ + \tilde{\delta}$ and $F_1^+ = F_0^+ + \delta'$ where $\tilde{\delta}, \delta' \in [0, 1]$.

To characterize $\gamma = \Gamma_B - \Gamma_I$, we first express $\Gamma_I$ and $\Gamma_B$ in terms of $\tilde{\delta}, \delta', T_0^+, T_1^+, F_0^+$, and $F_1^+$, and $Y_B|A$. Recall from Definition 3 that we define $\Gamma_I := \Pr\left[\hat{Y}_I = 1 \mid A = 0\right] - \Pr\left[\hat{Y}_I = 1 \mid A = 1\right]$.

We use Bayes' formula to re-write $\Pr\left[\hat{Y}_I = 1 \mid A = a\right], a \in A = \{0, 1\}$ in terms of the aforementioned variables. For $a \in \{0, 1\}$:

$$\Pr\left[\hat{Y}_I = 1 \mid A = a\right] = \Pr\left[\hat{Y}_B = 1 \mid A = a\right] \Pr\left[\hat{Y}_I = 1 \mid \hat{Y}_B = 1, A = a\right]$$
$$+ \Pr\left[\hat{Y}_B = 0 \mid A = a\right] \Pr\left[\hat{Y}_I = 1 \mid \hat{Y}_B = 0, A = a\right]$$
$$= \Pr\left[\hat{Y}_B = 1 \mid A = a\right] \cdot T_a^+ + \Pr\left[\hat{Y}_B = 0 \mid A = a\right] \cdot F_a^+ \tag{14}$$

Given our derivation above, we continue by substituting terms into $\Gamma_I$:

$$\Gamma_I = \Pr\left[\hat{Y}_B = 1 \mid A = 0\right] \cdot T_0^+ + \Pr\left[\hat{Y}_B = 0 \mid A = 0\right] \cdot F_0^+$$
$$- \Pr\left[\hat{Y}_B = 1 \mid A = 1\right] \cdot T_1^+ - \Pr\left[\hat{Y}_B = 0 \mid A = 1\right] \cdot F_1^+$$
$$= \left(T_1^+ + \tilde{\delta}\right) \Pr\left[\hat{Y}_B = 1 \mid A = 0\right] + \left(F_1^+ + \delta'\right) \Pr\left[\hat{Y}_B = 0 \mid A = 0\right]$$
$$- T_1^+ \Pr\left[\hat{Y}_B = 1 \mid A = 1\right] - F_1^+ \Pr\left[\hat{Y}_B = 0 \mid A = 1\right]$$
$$= T_1^+ \left(\Pr\left[\hat{Y}_B = 1 \mid A = 0\right] - \Pr\left[\hat{Y}_B = 1 \mid A = 1\right]\right)$$
$$+ F_1^+ \left(\Pr\left[\hat{Y}_B = 0 \mid A = 0\right] - \Pr\left[\hat{Y}_B = 0 \mid A = 1\right]\right)$$
$$+ \tilde{\delta} \Pr\left[\hat{Y}_B = 1 \mid A = 0\right] + \delta' \Pr\left[\hat{Y}_B = 0 \mid A = 0\right]$$
$$= T_1^+ \Gamma_B - F_1^+ \Gamma_B + \tilde{\delta} \Pr\left[\hat{Y}_B = 1 \mid A = 0\right] + \delta' \left(1 - \Pr\left[\hat{Y}_B = 1 \mid A = 0\right]\right)$$
$$= \Gamma_B \left(T_1^+ - F_1^+\right) + \left(\tilde{\delta} - \delta'\right) \Pr\left[\hat{Y}_B = 1 \mid A = 0\right] + \delta'$$

Now that we have derived $\Gamma_I$ in terms of our desired terms and $\Gamma_B$, we eliminate $\Gamma_I$ from the equation for $\gamma$:

$$\gamma = \Gamma_B - \Gamma_I$$
$$= \Gamma_B - \left(\Gamma_B(T_1^+ - F_1^+) + (\tilde{\delta} - \delta') \Pr\left[\hat{Y}_B = 1 \mid A = 0\right] + \delta'\right)$$
$$= \Gamma_B(1 + F_1^+ - T_1^+) + (\delta' - \tilde{\delta}) \Pr\left[\hat{Y}_B = 1 \mid A = 0\right] - \delta'$$
$$= \Gamma_B(1 + F_1^+ - T_1^+) - \delta' \cdot \left(1 - \Pr\left[\hat{Y}_B = 1 \mid A = 0\right]\right) - \tilde{\delta} \Pr\left[\hat{Y}_B = 1 \mid A = 0\right]$$
$$= \Gamma_B(1 + F_1^+ - T_1^+) - \delta' \Pr\left[\hat{Y}_B = 0 \mid A = 0\right] - \tilde{\delta} \Pr\left[\hat{Y}_B = 1 \mid A = 0\right]. \tag{15}$$

$\square$

**Corollary 1.** *Let $T_1^+, F_1^+$ as defined in Section 3. Then:*

$$1 + F_1^+ - T_1^+ > 0.$$

*Proof.* Note that $F_1^+, T_1^+ \in [0, 1]$, therefore $1 + F_1^+ - T_1^+ \geq 0$. We will show $1 + F_1^+ - T_1^+ = 0$ if and only if $T^+ = 1$ and $F^+ = 0$, *i.e.* if $I(\cdot)$ has perfect fidelity w.r.t. $B(\cdot)$. We re-arrange

Table 1: The accuracy (Acc.) and demographic parity gap (Gap) of the black-box models–AdaBoost, Deep Neural Networks (DNN), Random Forests (RF) and Gradient Boosted Decision Trees (XgBoost) evaluated on the suing set.

| Dataset | AdaBoost | | DNN | | RF | | XgBoost | |
|---------|------|------|------|------|------|------|------|------|
| | Acc. | Gap | Acc. | Gap | Acc. | Gap | Acc. | Gap |
| COMPAS | 0.68 | 0.21 | 0.68 | 0.25 | 0.67 | 0.25 | 0.68 | 0.27 |
| Adult Income | 0.85 | 0.12 | 0.85 | 0.16 | 0.86 | 0.17 | 0.86 | 0.17 |
| Bank Marketing | 0.91 | 0.07 | 0.91 | 0.09 | 0.91 | 0.10 | 0.91 | 0.10 |

$1 + F_1^+ - T_1^+ = 0$ as:

$$T_1^+ - F_1^+ = 1$$

Since $F_1^+, T_1^+ \in [0, 1]$ and the size of interval $[0, 1]$ is $1 - 0 = 1 = T_1^+ - F_1^+$, $T_1^+$ and $F_1^+$ must be endpoints 1 and 0, respectively. □

# E  Details on Experimental Setup of FRAUD-Detect

In this section, we provide additional details on our experimental setup.

**Datasets.** COMPAS [5] is a recidivism dataset released by ProPublica that includes data on $6,167$ offenders in prison from Broward County (Florida), with the classification task consisting in predicting whether an inmate will re-offend in the two years after their release.[3] We consider race (African-American, Caucasian) as the sensitive attribute in COMPAS. Adult Income [22] includes 48,842 profiles of individuals, each characterized by 13 attributes, drawn from the U.S census. The Adult Income binary classification task is to predict whether an individual's income will exceed $50,000 a year. We use gender (Female, Male) as the sensitive attribute in Adult Income. Finally, Bank Marketing [34] constitutes information collected from a direct marketing campaign of a Portuguese banking institution between 2008 to 2013. This dataset contains 41,188 profiles of individuals, each described by 20 attributes with the classification task being the prediction of subscription to a term deposit. We consider age (between 30-60 or not between 30-60) as the sensitive attribute in Bank Marketing.

**Training of black-box and interpretable models.** For each experiment, the dataset is split randomly into three partitions: the training set (67%), the suing set (16.5%) and the testing set (16.5%), following the same procedure as in [3]. Each experiment is repeated with 10 separate random seeds. The black-box models are learned from the training data points with their associated true labels. However, the interpretable models are trained from the suing set, which has been labelled by the black-box model. More precisely, we use the exponentiated gradient algorithm [1] subject to the fairwashing constraint from the FairLearn Library [9] to learn the fairwashed interpretable model. The exponentiated gradient algorithm when applied for fairness identifies the saddle point solution jointly maximizing the fairness and minimizing the loss. To learn the fairwashed interpretable model of the informed adversary, we extend the Fairness In The Rashomon Set (FaiRS) [19] framework by adding an additional constraint on $\mathcal{C}_{KL}$. FaiRS reduces the problem of minimizing the unfairness under two constraints on the fidelity and $\mathcal{C}_{KL}$ to a saddle point problem, which is solved with the exponentiated gradient technique [1]. Table 1 summarizes the performance of the considered black-box models on each suing set.

**Implementation.** FRAUD-Detect solely requires the confusion matrices from the interpretable model for both the majority and minority group. These matrices are flattened, then compared via KL as outlined in Equation 10. Experiments were conducted on Intel Xeon Silver 4110 CPUs (2.10 GHz, 8GB RAM) and the KL divergence detector implemented in Python 3.8 using scikit-learn [36]. Given that the KL divergence constraint is not a linear constraint on the confusion matrices, the extended version of FaiRS may fail to obtain the global minimum of the demographic parity gap on the set

---

[3]We note that considerable bias and confounding exists in the true labels for recidivism datasets at large. We do not necessarily condone use of recidivism datasets. However, COMPAS exists as a standard fairness benchmark and thus, we use COMPAS to validate our method.

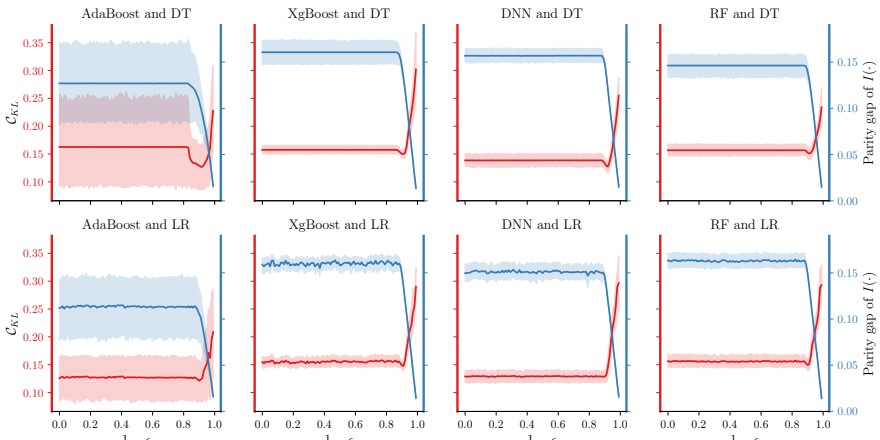

Figure 4: Kullback–Leibler divergence between confusion matrices of subpopulations, $\mathcal{C}_{\mathrm{KL}}$ and demographic parity gap as a function of $\epsilon$ fairwashing in interpretable models explaining black-box models using Adult Income dataset. We use Logistic Regression (LR) and Decision Tree (DT) as interpretable models, while AdaBoost, Deep Neural Network (DNN), Random Forest (RF) and Gradient Boosted Decision Trees (XgBoost) are used as black-box models.

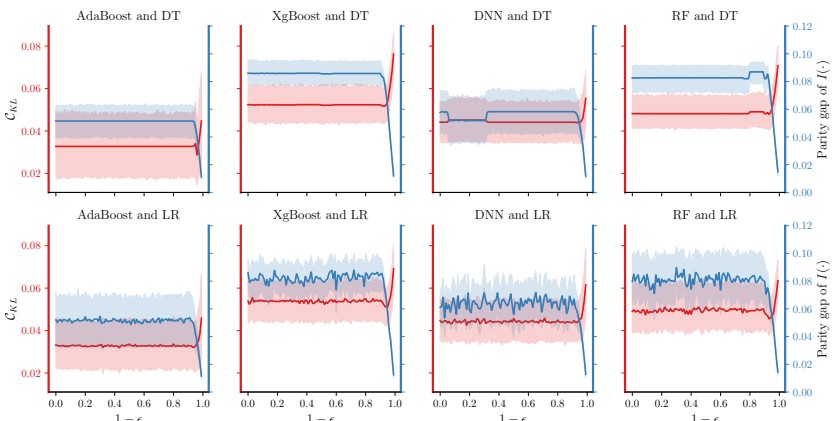

Figure 5: Kullback–Leibler divergence between confusion matrices of subpopulations, $\mathcal{C}_{\mathrm{KL}}$ and demographic parity gap as a function of $\epsilon$ fairwashing in interpretable models explaining black-box models using Marketing dataset. We use Logistic Regression (LR) and Decision Tree (DT) as interpretable models, while AdaBoost, Deep Neural Network (DNN), Random Forest (RF) and Gradient Boosted Decision Trees (XgBoost) are used as black-box models.

of high-fidelity interpretable models. In our experiments, our approach converges for two datasets, namely COMPAS and Marketing, but does not achieve the global minimum of the demographic parity gap for Adult Income. We are planning to address this in future work by focusing on detecting fairwashing using linear constraints.

## F    Additional results

**FRAUD-Detect successfully detects fairwashing.** Figures 4 and 5 illustrate both the demographic parity gap and $\mathcal{C}_{\mathrm{KL}}$ of interpretable models as a function of $\epsilon$ for two datasets (Adult income and Bank marketing), respectively.

**FRAUD-Detect is robust to the informed adversary.** Figures 6 and 7 show the range of the demographic parity gap of high-fidelity fairwashed interpretable models subjected to a constraint on the $\mathcal{C}_{\text{KL}}$ by solving the optimization problem of Equation (11). We use four types of black-box models (AdaBoost, XgBoost, DNN and Random Forest).

## G  Fairwashing Detection Thresholds

Table 2 reports several fairwashing detection thresholds observed in Figures 2, 4 and 5.

Table 2: $\mathcal{C}_{KL}$ values for each dataset, black-box model, and interpretable model. These values are chosen from average KL divergence values at 5% and 50% fairwashing.

| Dataset | IM | AdaBoost | | DNN | |
|---|---|---|---|---|---|
| | | 5% | 50% | 5% | 50% |
| COMPAS | DT | $0.105 \pm 0.035$ | $0.206 \pm 0.078$ | $0.136 \pm 0.037$ | $0.185 \pm 0.065$ |
| | LR | $0.101 \pm 0.029$ | $0.164 \pm 0.075$ | $0.143 \pm 0.044$ | $0.231 \pm 0.104$ |
| Adult Income | DT | $0.128 \pm 0.067$ | $0.148 \pm 0.086$ | $0.134 \pm 0.015$ | $0.154 \pm 0.029$ |
| | LR | $0.132 \pm 0.069$ | $0.170 \pm 0.117$ | $0.128 \pm 0.017$ | $0.213 \pm 0.048$ |
| Bank Marketing | DT | $0.035 \pm 0.022$ | $0.034 \pm 0.025$ | $0.045 \pm 0.017$ | $0.046 \pm 0.018$ |
| | LR | $0.033 \pm 0.018$ | $0.036 \pm 0.022$ | $0.042 \pm 0.015$ | $0.049 \pm 0.020$ |

| Dataset | IM | RF | | XgBoost | |
|---|---|---|---|---|---|
| | | 5% | 50% | 5% | 50% |
| COMPAS | DT | $0.149 \pm 0.040$ | $0.227 \pm 0.095$ | $0.170 \pm 0.041$ | $0.248 \pm 0.119$ |
| | LR | $0.145 \pm 0.043$ | $0.215 \pm 0.088$ | $0.164 \pm 0.044$ | $0.261 \pm 0.124$ |
| Adult Income | DT | $0.154 \pm 0.017$ | $0.170 \pm 0.024$ | $0.153 \pm 0.011$ | $0.196 \pm 0.036$ |
| | LR | $0.151 \pm 0.016$ | $0.207 \pm 0.044$ | $0.148 \pm 0.012$ | $0.200 \pm 0.041$ |
| Bank Marketing | DT | $0.048 \pm 0.013$ | $0.054 \pm 0.014$ | $0.053 \pm 0.015$ | $0.059 \pm 0.016$ |
| | LR | $0.045 \pm 0.014$ | $0.048 \pm 0.015$ | $0.052 \pm 0.014$ | $0.053 \pm 0.015$ |

## H  Per seed fine grain $C_{KL}$ versus fairness

Figures 8, 9, and 10 show the per-seed results aggregated to form figures 2, 4, and 5, respectively. In these figures (especially in 8 and 10), the order of increasing black-box fairnesses across the 10 seeds is often replicated in the order of decreasing interpretable model $\mathcal{C}_{\text{KL}}$ values across the same seeds; thus a seed with a low black-box fairness has higher $\mathcal{C}_{\text{KL}}$ values than a seed with high black-box fairness.

Some seeds from the logistic regression interpretable model (e.g. Figure 9 AdaBoost and logistic regression green and grey lines) show discontinuities at high epsilons where fairwashing is significant. There, the $\mathcal{C}_{\text{KL}}$ values jump to infinity due to a division by zero in the KL log term when there are 0s in the minority subpopulation confusion matrix.

We also observe that for several seeds across all datasets and models, the fairness of the honest interpretable model is imperfectly aligned and usually higher than the black-box fairness. This is unintentional fairwashing, which may arise from the relatively small capacity of the interpretable model compared to the black-box model.

## I  Bounds for KL Divergence of the Confusion Matrices

First, denote $\gamma_1 = \Gamma_B(1 + F_1^+ - T_1^+) = \Gamma_B(F_1^- + F_1^+)$ and similarly $\gamma_0 = \Gamma_B(F_0^- + F_0^+)$. Depending on the context of the problem either subgroup could be the more sensitive subgroup, so in Equation (6) we implicitly assumed that subgroup is $z = 1$: $\gamma = \gamma_1$. Equation (10) defines the detector based on the flattened confusion matrix, but since for a binary classification problem, the

rank of the confusion matrix is 2, it is enough to consider $\text{KL}\left(\begin{bmatrix} F_0^+ \\ F_0^- \end{bmatrix}, \begin{bmatrix} F_1^+ \\ F_1^- \end{bmatrix}\right)$.[4] We will derive upper and lower bounds on this divergence measure in terms of $\gamma_0, \gamma_1, F_i^+, F_i^-; \ i \in \{0,1\}$. First, note that $\gamma_1$ is the sum of type I $F_1^-$ and type II $F_1^+$ detection errors; scaled by the unfairness of the black-box model $\Gamma_B$. In a detection problem, typically, we bound the type I error and minimize the type II error. The particular choice of type I and type II error is context-dependent (are false positives a bigger concern than false negatives or not?). In the context of fairwashing, a reasonable choice is that

$$F_1^+ \leq F_1^-, \tag{16}$$
$$F_0^+ \leq F_0^-$$

That is, the cost of falsely detecting a fairwashed model is larger than missing fairwashing—due to the legal and liability considerations, for instance.

We can bound the irreducible fairwashing gap $\gamma_i$:

$$2\Gamma_b F_1^+ \leq \gamma_1 \leq 2\Gamma_b F_1^-, \tag{17}$$
$$2\Gamma_b F_0^+ \leq \gamma_0 \leq 2\Gamma_b F_0^-$$

Using this we bound the aforementioned KL divergence:

$$\text{KL}\left(\begin{bmatrix} F_0^+ \\ F_0^- \end{bmatrix}, \begin{bmatrix} F_1^+ \\ F_1^- \end{bmatrix}\right) = F_0^+ \log\left(\frac{F_0^+}{F_1^+}\right) + F_0^- \log\left(\frac{F_0^-}{F_1^-}\right) \tag{18}$$

$$\leq F_0^- \log\left(\frac{F_0^+}{F_1^+} \cdot \frac{F_0^-}{F_1^-}\right) \tag{19}$$

$$= F_0^- \log\left(\frac{F_0^+}{F_1^-} \cdot \frac{F_0^-}{F_1^+}\right) \tag{20}$$

$$\leq F_0^- \log\left(\frac{\gamma_0/2\Gamma_B}{\gamma_1/2\Gamma_B} \cdot \frac{F_0^-}{F_1^+}\right) \tag{21}$$

$$= F_0^- \log\left(\frac{\gamma_0}{\gamma_1} \cdot \frac{F_0^-}{F_1^+}\right) \tag{22}$$

Similarly,

$$\text{KL}\left(\begin{bmatrix} F_0^+ \\ F_0^- \end{bmatrix}, \begin{bmatrix} F_1^+ \\ F_1^- \end{bmatrix}\right) \geq F_0^+ \log\left(\frac{F_0^+}{F_1^+} \cdot \frac{F_0^-}{F_1^-}\right) \tag{23}$$

$$= F_0^+ \log\left(\frac{F_0^+}{F_1^-} \cdot \frac{F_0^-}{F_1^+}\right) \tag{24}$$

$$\geq F_0^+ \log\left(\frac{F_0^+}{F_1^-} \cdot \frac{\gamma_0/2\Gamma_B}{\gamma_1/2\Gamma_B}\right) \tag{25}$$

$$= F_0^+ \log\left(\frac{F_0^+}{F_1^-} \cdot \frac{\gamma_0}{\gamma_1}\right) \tag{26}$$

Therefore, $\kappa := \text{KL}\left(\begin{bmatrix} F_0^+ \\ F_0^- \end{bmatrix}, \begin{bmatrix} F_1^+ \\ F_1^- \end{bmatrix}\right)$ is bounded from below and above by:

$$F_0^+ \log\left(\frac{F_0^+}{F_1^-} \cdot \frac{\gamma_0}{\gamma_1}\right) \leq \kappa \leq F_0^- \log\left(\frac{\gamma_0}{\gamma_1} \cdot \frac{F_0^-}{F_1^+}\right) \tag{27}$$

---

[4]That is to say, any two rates from the confusion matrix is enough to construct the full matrix. We note that in this section, we work with false positive rate $F^+$ and false negative rate $F^-$ to simplify the presentation. The rest of the paper uses the entire confusion matrix which as we have pointed out in Theorem 1 is not necessary but is also harmless.

Equivalently, we can re-write these bounds in terms of $F^+$ and $T^+$ to match Theorem 1 (using the fact that $F^- = 1 - T^+$):

$$F_0^+ \log \left( \frac{F_0^+}{1 - T_0^+} \cdot \frac{\gamma_0}{\gamma_1} \right) \leq \kappa \leq (1 - T_0^+) \log \left( \frac{\gamma_0}{\gamma_1} \cdot \frac{1 - T_0^+}{F_1^+} \right) \tag{28}$$

Therefore any choice between $\kappa_{min} := F_0^+ \log \left( \frac{F_0^+}{F_1^-} \cdot \frac{\gamma_0}{\gamma_1} \right)$ and $\kappa_{max} := F_0^- \log \left( \frac{\gamma_0}{\gamma_1} \cdot \frac{F_0^-}{F_1^+} \right)$ is valid. $\kappa = \kappa_{min}$ is the most conservative choice which increases the risk of false discovery (of fairwashing), while $\kappa = \kappa_{max}$ is a permissive choice that minimizes the risk of false discovery at the expense of increase in the false negatives (not detecting fairwashing). A balanced choice for threshold could be the intermediate point:

$$\bar{\kappa} = \frac{1}{2} \left( F_0^+ \log \left( \frac{F_0^+}{F_1^-} \cdot \frac{\gamma_0}{\gamma_1} \right) + F_0^- \log \left( \frac{\gamma_0}{\gamma_1} \cdot \frac{F_0^-}{F_1^+} \right) \right) \tag{29}$$

$$= \frac{1}{2} \log \left( \left( \frac{F_0^+}{F_1^-} \right)^{F_0^+} \left( \frac{F_0^-}{F_1^+} \right)^{F_0^-} \left( \frac{\gamma_0}{\gamma_1} \right)^{F_0^+ + F_0^-} \right) \tag{30}$$

$$= \frac{1}{2} \log \left( \left( \frac{F_0^+}{F_1^-} \right)^{F_0^+} \left( \frac{F_0^-}{F_1^+} \right)^{F_0^-} \left( \frac{\gamma_0}{\gamma_1} \right)^{\gamma_0/\Gamma_B} \right) \tag{31}$$

$$= \frac{1}{2} \log \left( \left( \frac{F_0^+}{F_1^-} \right)^{F_0^+} \left( \frac{F_0^-}{F_1^+} \right)^{F_0^- - \gamma_0/\Gamma_B} \left( \frac{\gamma_0}{\gamma_1} \right)^{\gamma_0/\Gamma_B} \right) \tag{32}$$

$$= \frac{1}{2} \log \left( \left( \frac{F_0^+ F_0^-}{F_1^- F_1^+} \right)^{F_0^+} \left( \frac{F_0^-}{F_1^+} \right)^{-\gamma_0/\Gamma_B} \left( \frac{\gamma_0}{\gamma_1} \right)^{\gamma_0/\Gamma_B} \right) \tag{33}$$

$$= \frac{1}{2} \log \left( \left( \frac{F_0^+}{F_1^+} \right)^{F_0^+} \left( \frac{F_0^-}{F_1^-} \right)^{F_0^+} \left( \frac{F_0^-}{F_1^+} \right)^{-\gamma_0/\Gamma_B} \left( \frac{\gamma_0}{\gamma_1} \right)^{\gamma_0/\Gamma_B} \right) \tag{34}$$

$$= \frac{1}{2} \log \left( \left( \frac{F_0^+}{F_1^+} \right)^{F_0^+} \left( \frac{F_0^-}{F_1^-} \right)^{F_0^+} \left( \frac{\gamma_1}{\Gamma_B} - \frac{F_1^-}{F_0^-} \right)^{\gamma_0/\Gamma_B} \left( \frac{\gamma_0}{\gamma_1} \right)^{\gamma_0/\Gamma_B} \right) \tag{35}$$

$$= \frac{F_0}{2} \left( \Delta_l(F^+) + \Delta_l(F^-) \right) + \frac{\gamma_0}{2\Gamma_B} \log \left( \frac{\gamma_0}{\Gamma_B} - \frac{\gamma_0}{\gamma_1} \frac{F_1^-}{F_0^-} \right) \tag{36}$$

$$= \frac{F_0}{2} \left( \Delta_l(F^+) + \Delta_l(F^-) \right) + \frac{\gamma_0}{2\Gamma_B} \log \left( \frac{\gamma_0}{\Gamma_B} \right) + \frac{\gamma_0}{2\Gamma_B} \log \left( 1 - \frac{\Gamma_B}{\gamma_1} \frac{F_1^-}{F_0^-} \right) \tag{37}$$

$$= \frac{F_0}{2} \left( \Delta_l(F^+) + \Delta_l(F^-) \right) + \frac{\gamma_0}{2\Gamma_B} \log \left( \frac{\gamma_0}{\Gamma_B} \right) - \frac{\gamma_0}{2\Gamma_B} \left( \frac{\Gamma_B}{\gamma_1} \frac{F_1^-}{F_0^-} \right) - O \left( \frac{\Gamma_B^2}{\gamma_1^2} \left( \frac{F_1^-}{F_0^-} \right)^2 \right) \tag{38}$$

$$= \frac{F_0}{2} \left( \Delta_l(F^+) + \Delta_l(F^-) \right) + \frac{\gamma_0}{2\Gamma_B} \log \left( \frac{\gamma_0}{\Gamma_B} \right) - \frac{1}{2} \frac{\gamma_0}{\gamma_1} \left( \frac{F_1^-}{F_0^-} \right) - O \left( \frac{\Gamma_B^2}{\gamma_1^2} \left( \frac{F_1^-}{F_0^-} \right)^2 \right) \tag{39}$$

$$\bar{\kappa} = \frac{F_0}{2} \left( \Delta_l(F^+) + \Delta_l(F^-) \right) + \frac{\gamma_0}{2\Gamma_B} \log \left( \frac{\gamma_0}{\Gamma_B} \right) - \frac{1}{2} \frac{\gamma_0}{\gamma_1} \frac{1}{\Delta(F^-)} - O \left( \frac{\Gamma_B^2}{\gamma_1^2} \left( \frac{F_1^-}{F_0^-} \right)^2 \right) \tag{40}$$

where $\Delta(R) = R_0/R_1$ and $\Delta_l(R) = \log \Delta(R)$. In the last equation, we have used the taylor expansion of $\log(1 - x)$ at $x = 0$.

## J  Experimental results with two independent variables

In this section, we repeat experiments shown in Section 6 with the following adjustment:

$$\mathcal{C}_{KL} = \text{KL}([T_0^+, F_0^+], [T_1^+, F_1^+]),$$

in which we use only two independent variables of true-positive rate $T^+$ and false-positive rate $F^+$. Figures 11, 12 and 13 show similar results as Figures 2, 4 and 5.

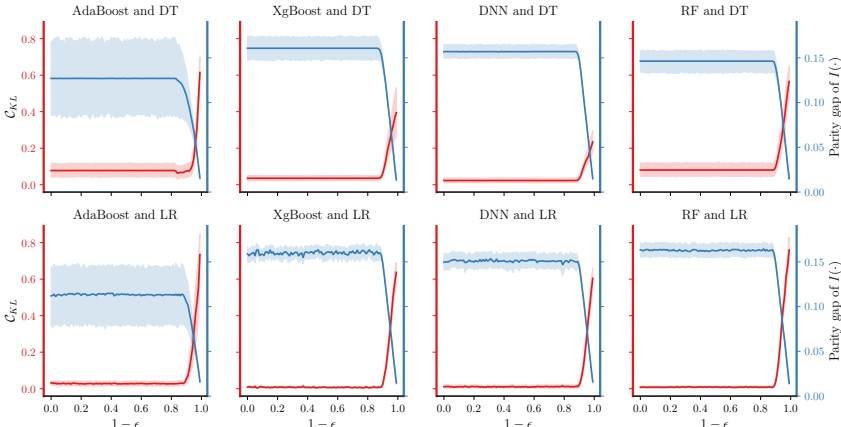

Figure 12: KL among altered confusion matrices of subgroups versus demographic parity unfairness of two explainable models, namely Logistic Regression (LR) and Decision Tree (DT), approximating four black-box models, namely AdaBoost, Deep Neural Network (DNN), Random Forest (RF), and Gradient Boosted Decision Trees (XgBoost) using Adult dataset. We consider 10 split of the dataset to the training set for the black-box model and suing set for the interpretable model.

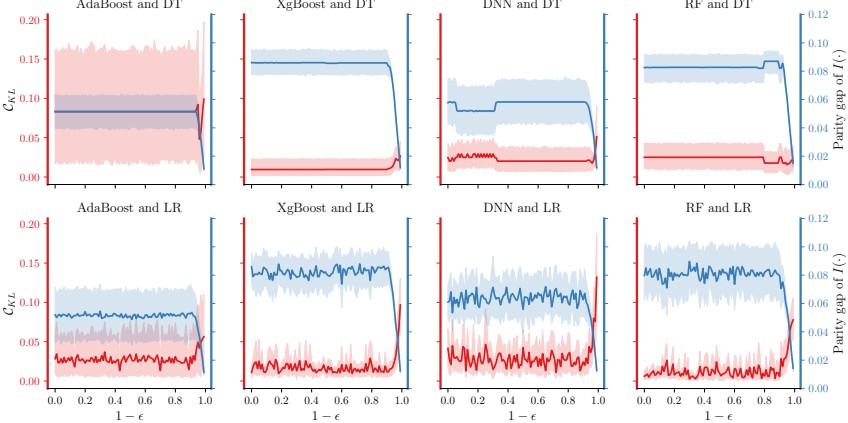

Figure 13: KL among altered confusion matrices of subgroups versus demographic parity unfairness of two explainable models, namely Logistic Regression (LR) and Decision Tree (DT), approximating four black-box models, namely AdaBoost, Deep Neural Network (DNN), Random Forest (RF), and Gradient Boosted Decision Trees (XgBoost), using Marketing dataset. We consider 10 split of the dataset to the training set for the black-box model and suing set for the interpretable model.

## K  Perfect Fairwashing is impossible

**Theorem 2.** *Consider binary-valued random variables $\hat{Y}_B$ black-box model predictions, $\hat{Y}_I$ proxy (interpretable) model predictions, and $A$ a sensitive attribute; then these conditions cannot be satisfied simultaneously:*

*1. Interpretable Model Demographic Parity*

$$\Pr\left[\hat{Y}_I = 1 \mid A = 0\right] = \left[\hat{Y}_I = 1 \mid A = 1\right]$$

,

*2. Non-detectable*

$$\Pr\left[Y_I = 1 \mid Y_B = 1, A = 0\right] = \Pr\left[Y_I = 1 \mid Y_B = 1, A = 1\right] := T^+ \quad (41)$$

$$\Pr\left[Y_I = 1 \mid Y_B = 0, A = 0\right] = \Pr\left[Y_I = 1 \mid Y_B = 0, A = 1\right] := F^+. \quad (42)$$

*Proof.* Let Equation (41) be equal to $T^+$, and Equation (42) to $F^+$. Define the base rates of the black-box model on subpopulations 0 and 1 as:

$$\mu_0 = \Pr[\hat{Y}_B = 1 | A = 0]$$

$$\mu_1 = \Pr[\hat{Y}_B = 1 | A = 1].$$

Using the rule of total probabilities, for $a \in [0, 1]$,

$$\Pr\left[\hat{Y}_I = 1 \mid A = a\right] = \Pr[\hat{Y}_B = 1 | A = a] \Pr\left[\hat{Y}_I = 1 \mid \hat{Y}_B = 1, A = a\right]$$

$$+ \Pr[\hat{Y}_B = 0 | A = a] \Pr\left[\hat{Y}_I = 1 \mid \hat{Y}_B = 0, A = a\right] \quad (43)$$

$$= \mu_a \cdot T^+ + (1 - \mu_a) \cdot F^+ \quad (44)$$

Fairness according to parity requires that

$$\Pr\left[\hat{Y}_I = 1 \mid A = 0\right] \overset{?}{=} \Pr\left[\hat{Y}_I = 1 \mid A = 1\right]. \quad (45)$$

Substituting Equation (43) and Equation (45) and re-organizing we have:

$$(\mu_0 - \mu_1) \cdot T^+ + ((1 - \mu_0) - (1 - \mu_1)) \cdot F^+ \overset{?}{=} 0 \quad (46)$$

**Case 1:** If $T^+ = F^+$, by eliminating both and recognizing that $(1 - \mu_0) = 1 - \mu_0$, we see that the equality Equation (45) reduces to $0 = 0$ which is trivially correct.

**Case 2:** $T^+ \neq F^+$  Simplifying Equation (46) further and eliminating $T^+ - F^+ \neq 0$

$$\mu_0(T^+ - F^+) - \mu_1 \cdot (T^+ - F^+) \overset{?}{=} (T^+ - F^+)$$

$$\Leftrightarrow \quad \mu_0 \overset{?}{=} 1 + \mu_1.$$

But these are probabilities, for Equation (45) to hold we must have that

$$\mu_1 = 0 \quad (47)$$

$$\mu_0 = 1,$$

which means that the black-box predictions exclusively depend on the sensitive attribute. In other words, for demographic parity to hold for the interpretable model, the black-box models predictions all samples of one subpopulation $A = 0$ and reject the others $A = 0$. Clearly, this is unfair to one subpopulation.

To establish the impossibility result, we further show that such an assumption is equivalent to requiring completely inaccurate black-box models with maximum error rates.

Consider the "true" base rates for each subpopulation:

$$\Pr[Y = 1 \mid A = 0] = \Pr\left[Y = 1 \mid \hat{Y}_B = 1, A = 0\right]\mu_0 + \Pr\left[Y = 1 \mid \hat{Y}_B = 0, A = 0\right]\Pr\left[\hat{Y}_B = 0 \mid A = 0\right]$$

$$\Pr[Y = 1 \mid A = 1] = \Pr[Y = 1 \mid \hat{Y}_B = 1, A = 1]\mu_1 + \Pr\left[Y = 1 \mid \hat{Y}_B = 1, A = 1\right](1 - \mu_1)$$

If interpretable demographic parity Equation (47) holds then, these base rates reduce to:

$$\Pr[Y = 1 \mid A = 0] = \Pr\left[Y = 1 \mid \hat{Y}_B = 1, A = 0\right]$$

$$\Pr[Y = 1 \mid A = 1] = \Pr\left[Y = 1 \mid \hat{Y}_B = 0, A = 1\right] \implies Y \perp \hat{Y}_B \mid A \implies \hat{Y}_B \perp Y \mid A.$$

In other words, the black-box model predictions is completely independent of the true label and exclusively dependent on the sensitive attribute $A$. $\qquad\square$

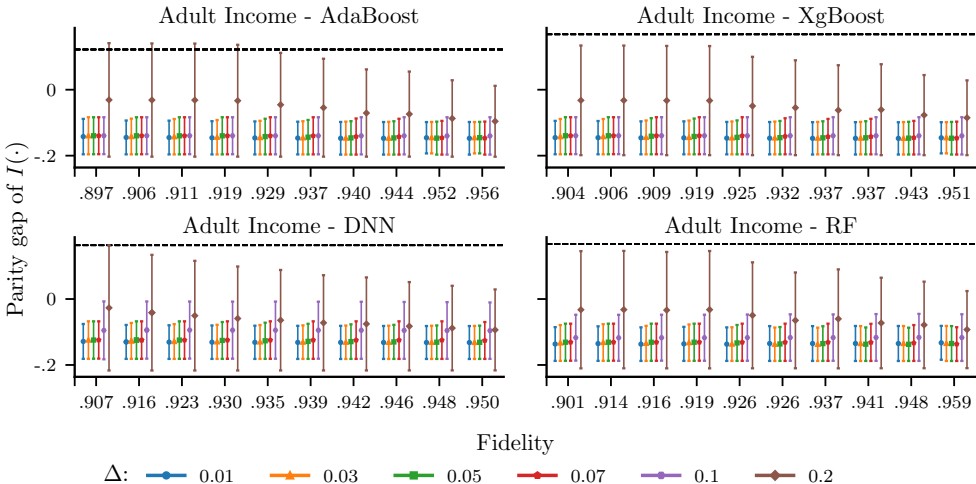

Figure 6: Range of the demographic parity gap of high-fidelity fairwashed logistic regression interpretable models subjected to a constraint on the $\mathcal{C}_{KL}$ ($\Delta \in \{0.01, 0.03, 0.05, 0.07, 0.10, 0.20\}$), explaining AdaBoost, Deep Neural Network (DNN), Random Forest (RF) and Gradient Boosted Decision Trees (XgBoost) black-box models trained on Adult Income dataset. Horizontal lines denote the demographic parity gap of the black-box models.

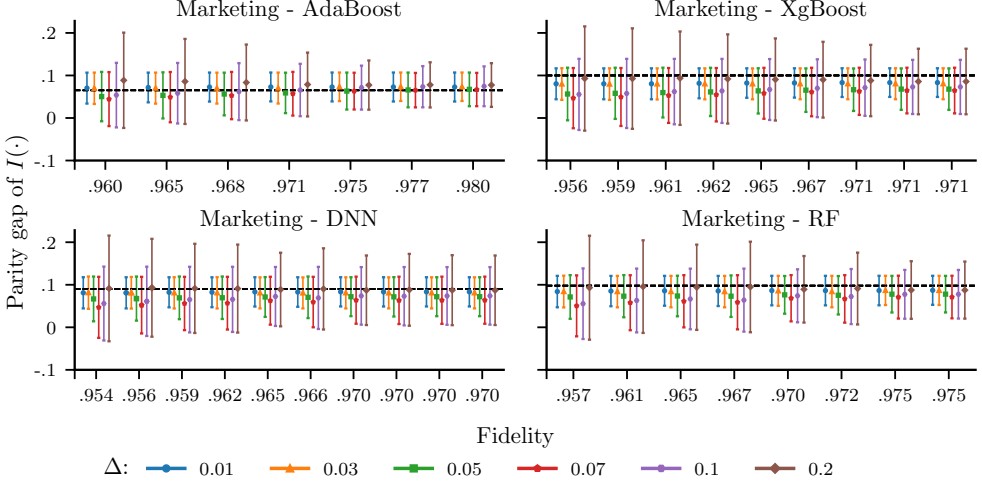

Figure 7: Range of the demographic parity gap of high-fidelity fairwashed logistic regression interpretable models subjected to a constraint on the $\mathcal{C}_{KL}$ ($\Delta \in \{0.01, 0.03, 0.05, 0.07, 0.10, 0.20\}$), explaining AdaBoost, Deep Neural Network (DNN), Random Forest (RF) and Gradient Boosted Decision Trees (XgBoost) black-box models trained on Marketing dataset. Horizontal lines denote the demographic parity gap of the black-box models.

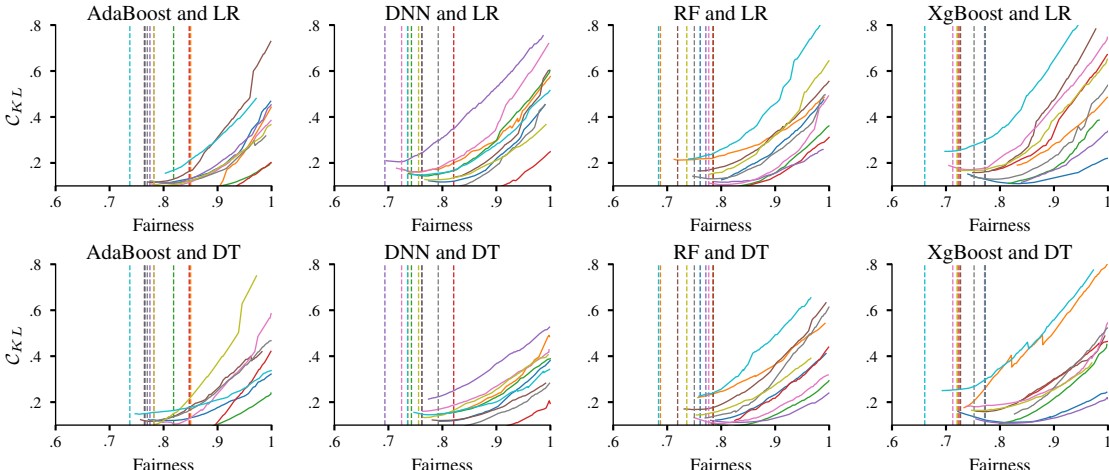

Figure 8: Kullback–Leibler divergence, $\mathcal{C}_{KL}$, between confusion matrices of subgroups versus demographic parity fairness of two interpretable models, namely Logistic Regression (LR) and Decision Tree (DT), approximating four black-box models, namely AdaBoost, Deep Neural Network (DNN), Random Forest (RF), and Gradient Boosted Decision Trees (XgBoost), using COMPAS dataset. We consider 10 split of the dataset to the training set for the black-box model and suing set for the interpretable model. Vertical dashed lines represent unfairness values of the black-box models.

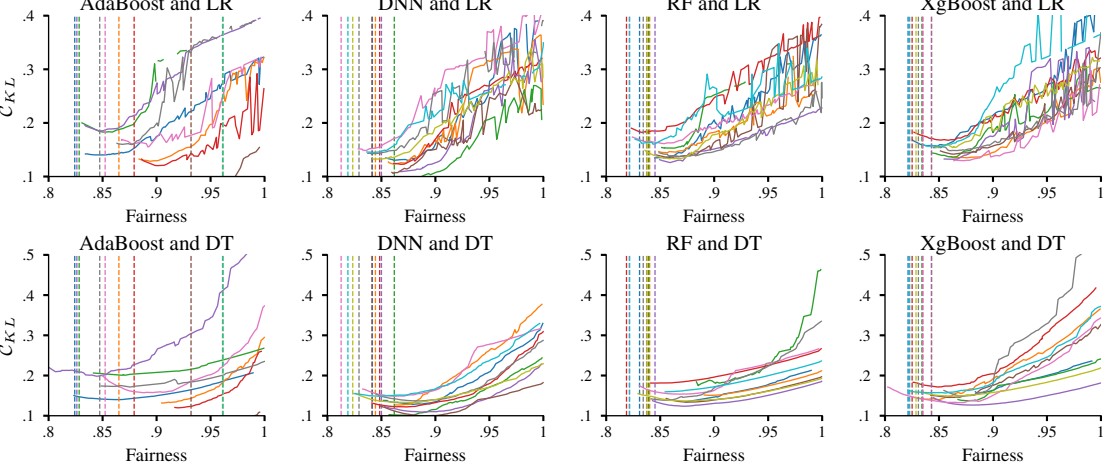

Figure 9: Kullback–Leibler divergence, $\mathcal{C}_{KL}$, between confusion matrices of subgroups versus demographic parity fairness of two interpretable models, namely Logistic Regression (LR) and Decision Tree (DT), approximating four black-box models, namely AdaBoost, Deep Neural Network (DNN), Random Forest (RF), and Gradient Boosted Decision Trees (XgBoost) using Adult dataset. We consider 10 split of the dataset to the training set for the black-box model and suing set for the interpretable model. Vertical dashed lines represent unfairness values of the black-box models.

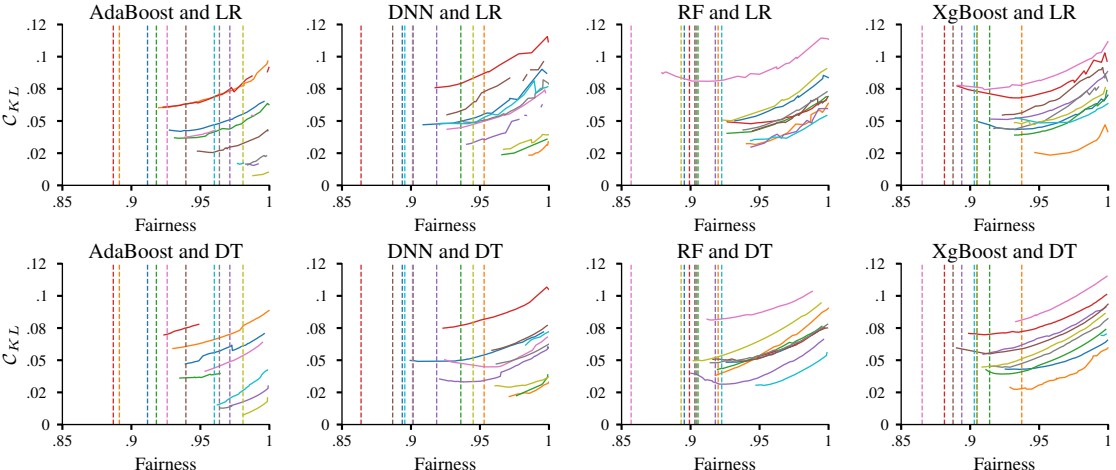

Figure 10: Kullback–Leibler divergence, $\mathcal{C}_{KL}$, between confusion matrices of subgroups versus demographic parity fairness of two interpretable models, namely Logistic Regression (LR) and Decision Tree (DT), approximating four black-box models, namely AdaBoost, Deep Neural Network (DNN), Random Forest (RF), and Gradient Boosted Decision Trees (XgBoost), using Marketing dataset. We consider 10 split of the dataset to the training set for the black-box model and suing set for the interpretable model. Vertical dashed lines represent unfairness values of the black-box models.

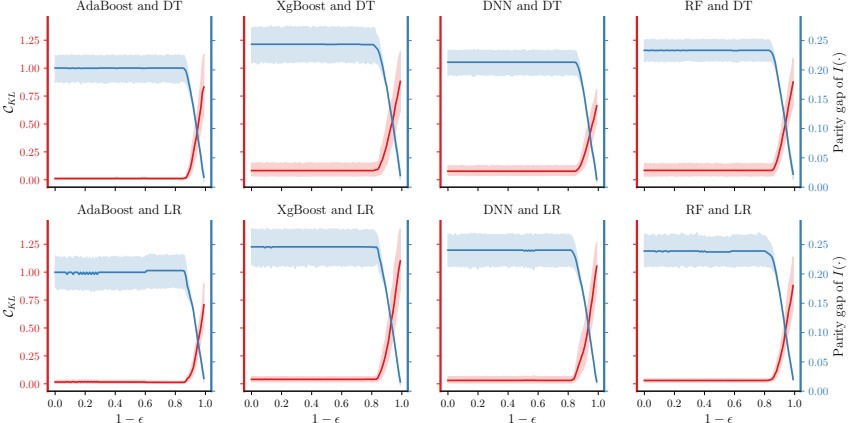

Figure 11: KL among altered confusion matrices of subgroups versus demographic parity unfairness of two explainable models, namely Logistic Regression (LR) and Decision Tree (DT), approximating four black-box models, namely AdaBoost, Deep Neural Network (DNN), Random Forest (RF), and Gradient Boosted Decision Trees (XgBoost), using COMPAS dataset. We consider 10 split of the dataset to the training set for the black-box model and suing set for the interpretable model.