# OpenReview forum: "Washing The Unwashable : On The (Im)possibility of Fairwashing Detection"
_NeurIPS.cc/2022/Conference — NeurIPS 2022 Accept_

### Official Review · Reviewer_WPPT · 2022-07-09

**Rating:** 7
**Confidence:** 4
**Soundness:** 3 good
**Presentation:** 4 excellent
**Contribution:** 3 good

**Summary:**

The paper characterizes the problem of fairwashing for interpretable models. It proposes a fairwashed model detection method based on measuring the difference over subpopulations in true-positive and false-positive rates of the interpretable model w.r.t. the black-box model. It shows the method’s sufficiency theoretically and effectiveness empirically even under the assumption of an informed attacker. The evaluation is conducted on two interpretable models of four classification models on three widely used fairness datasets.

**Questions:**

1.For other fairness criteria such as equality of opportunity, how easy is it to adapt the current method?

**Limitations:**

They are well addressed.

**Strengths And Weaknesses:**

## Strengths
1.The problem is well-motivated. Fairwashing can be a practical threat in reality.

2.The theoretical analyses look correct.

3.The proposed method is comprehensively evaluated and shows decent performance for practical usage. In particular, the setting of an informed adversary strengthens the generalizability of the proposed method.

4.The paper is well-written, well-organized, and easy to follow.

## Weaknesses
1.The method currently only applies to one fairness definition. However, in practice, other definitions like equality of opportunity are also commonly used.

---

> ### Author Response · Authors · 2022-08-02
> **Response to Reviewer WPPT**
>
> We thank the reviewer for the comments, and for highlighting that our paper is well-written, well-motivated, comprehensive in its experiments as well as for recognising the importance of the evaluation of our method against an informed adversary.  Below, we will answer your comment.
>
> > **The method currently only applies to one fairness definition. However, in practice, other definitions like equality of opportunity are also commonly used. For other fairness criteria such as equality of opportunity, how easy is it to adapt the current method?**
>
> We thank the reviewer for the suggestion. As discussed in Section 7, our main reason for focusing on demographic parity in this work is that it does not assume knowledge or even the existence of true labels (see Section 2.1). Nevertheless, following the reviewer’s suggestion, we have evaluated our detector on other fairness metrics. In particular, we have considered the equalized odds fairness metric which in addition to constraining the true positive rates (as would equality of opportunity) also constrains the false positive rates across groups. We choose the Adult dataset (black-box model: DNNs and interpretable model: Decision trees) as it displays an inherent equalized odds unfairness. The results below show that, similarly to our demographic parity results, we observe a relationship between the equalized odds gap of interpretable models and the associated $C_KL$ values: as fairwashing becomes more severe ( $1-\epsilon \downarrow$), the equalized odds decreases while $C_{KL}$ increases.
>
> |$1-\epsilon$:|0.6|0.8|0.9|0.95|0.99|
> |-------|--------|--------|--------|--------|--------|
> |Eodds:|.077345|.07616|.06650|.04473|.02191|
> |$C_{KL}$:|.12300|.13329|.13883|.14118|.14131|

---

> > ### Comment · Reviewer_WPPT · 2022-08-05
> > **Thank you for the response**
> >
> > I want to thank the authors for the response which addresses my question. I hope the authors can include this additional results on equalized odds in the updated appendix (probably as an extra section).

---

> > > ### Author Response · Authors · 2022-08-09
> > > **Thank you!**
> > >
> > > Thank you very much for your time reading our responses. Sure we will add these additional results in the appendix of the final version of our manuscript.

---

### Official Review · Reviewer_53g2 · 2022-07-11

**Rating:** 6
**Confidence:** 3
**Soundness:** 3 good
**Presentation:** 3 good
**Contribution:** 2 fair

**Summary:**

This paper presents a theoretical analysis of fair washing and demonstrates that completely reducing fair washing in certain cases is not possible. In addition, they present a simple technique that looks at the KL between surrogate model confusing matrices to detect fairwashing. They evaluate their technique and find it helps detect fairwashing issues.

**Questions:**

- Figure 3 is a bit hard to follow. Could you help clarify this figure? Where does the dotted line come from? Why are there multiple fidelity values for every $\Delta$?
- Could you help clarify the problem setup a bit more, per the weaknesses section?
- Could you clarify what you mean by interpretable model?


**Limitations:**

The authors have done a sufficient job.

**Strengths And Weaknesses:**

Strengths:
- Interesting problem
- The empirical evaluation of fraud detect is comprehensive, considering many different models and settings.

Weaknesses:
The main weakness is that the theory is missing explicitness in several places and there are gaps in what the exact setting is. Specifically, here are the main gaps I see:
- The initial problem setup is not defined explicitly enough. It seems like the setting is a classifier with a sensitive attribute, but the specifics here are lacking. For example, in equation (1), is this a multiclass or binary class setting? Is the protected attributed discrete or continuous? If it's continuous, is it allowed to take on multiple values? These details are a bit hazy right now making it hard to figure out where the claims apply exactly.
- What are the specifications of the interpretable model? Does this model mirror the predictions across multiple classes or just a single class?
- The construction of interpretable model $I$ is a bit hard to follow as well, and $I$ is introduced as just "a simple interpretable model I". I think what is exactly meant by the interpretable model needs to be stated more explicitly, because right now, it is a bit hazy and hard to follow. This makes it difficult to follow the statement of Theorem 1, where the theorem specifices and interpretable model, without this really being defined.

While the empirical results about the detection algorithm are pretty complete, it seems tricky to choose this threshold $\Delta$ that determines whether fair washing is going on. It would useful to provide more guidance here.

Overall, I think this paper needs to much more rigorously provide the problem setting and explicit definitions of what is going on.

---

> ### Author Response · Authors · 2022-08-02
> **Response to Reviewer 53g2**
>
> We thank the reviewer for the comments and for highlighting that our paper is interesting and comprehensive in its experiments.
>
>
> > **The initial problem setup is not defined explicitly enough. It seems like the setting is a classifier with a sensitive attribute, but the specifics here are lacking. For example, in equation (1), is this a multiclass or binary class setting? Is the protected attributed discrete or continuous? If it's continuous, is it allowed to take on multiple values? These details are a bit hazy right now making it hard to figure out where the claims apply exactly.**
>
> Following the standard assumptions made in the fairwashing literature ([ICML-2019] and [NeurIPS-2021]), we consider binary classification tasks, and binary-valued sensitive attributes. We clarified and formalized this in the revised version of the paper in Section 2.2 in which we introduce the black-box model and sensitive attributes. We have uploaded the revised version of the paper.
>
> -[ICML-2019] Ulrich Aïvodji, Hiromi Arai, Olivier Fortineau, Sébastien Gambs, Satoshi Hara, Alain Tapp: Fairwashing: the risk of rationalization. ICML 2019: 161-170.
>
> -[NeurIPS-2021] Ulrich Aïvodji, Hiromi Arai, Sébastien Gambs, Satoshi Hara: Characterizing the risk of fairwashing. NeurIPS 2021: 14822-14834.
>
> Our analysis can be extended to multi-class and multi-valued sensitive attributes but as written in Section 7 we leave this as future work, as it involves multi-class calibration results that are out of the scope of the current paper.
>
> > **What are the specifications of the interpretable model? Does this model mirror the predictions across multiple classes or just a single class? Could you clarify what you mean by interpretable model? The construction of interpretable model I is a bit hard to follow as well, and I is introduced as just "a simple interpretable model I". I think what is exactly meant by the interpretable model needs to be stated more explicitly, because right now, it is a bit hazy and hard to follow**
> > **This makes it difficult to follow the statement of Theorem 1, where the theorem specifices and interpretable model, without this really being defined.**
>
> We thank the reviewer for attention to detail. Indeed, Theorem 1 is quite broad because it is model-agnostic. Consider two machine learning models trained on the same dataset. One model is deemed to be more “interpretable.” This could be because of its tree-like construction that facilitates observing a decision boundary, or it could be a linear model with weights that signify the importance of each feature. The other model, deemed the black-box model, does not necessarily enjoy these characteristics—but instead may be more performant.
>
> Regardless of the particular architecture of models, Theorem 1 says that as long as we can characterize the false positive rate and true positive rate of the interpretable model’s decisions (with the reference points defined as black-box model decisions); we can fully characterize the fairwashing gap provided we know the unfairness of the black-box model.  Furthemore, the theorem shows that this gap cannot be reduced to zero and provides the lower bound $\gamma = \Gamma_B (1 + F^+_1 - T^+_1)$.

---

> > ### Author Response · Authors · 2022-08-02
> > **Response to Reviewer 53g2**
> >
> > > **While the empirical results about the detection algorithm are pretty complete, it seems tricky to choose this threshold Δ that determines whether fair washing is going on. It would useful to provide more guidance here.**
> >
> >
> > We thank the reviewer for the insightful comment. The question of choosing a threshold is a pertinent one. We provide two answers, one based on the theory we have presented in Section 3, and an empirical solution.
> >
> > ### Theory Solution
> > In Appendix I, we provide the following bounds for the KL divergence measure of FRAUD-detect:
> >
> > $\kappa := \operatorname{KL}\left(
> >         \begin{bmatrix} F^+_0 \\
> >                         F^-_0
> >         \end{bmatrix},
> >         \begin{bmatrix} F^+_1 \\
> >                         F^-_1
> >         \end{bmatrix}
> >         \right)$
> >
> > is bounded from below and above by:
> >
> > $
> >   F_0^+ \log\left(\frac{F_0^+}{F_1^-} \cdot \frac{\gamma_0}{\gamma_1}\right)
> >                         \leq  \kappa \leq
> >   F_0^- \log\left(\frac{\gamma_0}{\gamma_1} \cdot \frac{F_0^-}{F_1^+}\right)
> > $
> >
> > Any choice between
> > $\kappa_{min} := F_0^+ \log\left(\frac{F_0^+}{F_1^-} \cdot \frac{\gamma_0}{\gamma_1}\right)$
> > and
> > $\kappa_{max} := F_0^- \log\left(\frac{\gamma_0}{\gamma_1} \cdot \frac{F_0^-}{F_1^+}\right)$ is valid.
> > $\kappa = \kappa_{min}$ is the most conservative choice which increases the risk of false discovery (of fairwashing), while $\kappa = \kappa_{max}$ is a permissive choice that minimizes the risk of false discovery at the expense of increase in the false negatives (not detecting fairwashing). A balanced choice for threshold  could be the intermediate point:
> >
> > $\bar{\kappa}=\frac{F_{0}}{2}\left(\Delta_{l}\left(F^{+}\right)+\Delta_{l}\left(F^{-}\right)\right)+\frac{\gamma_{0}}{2 \Gamma_{B}} \log \left(\frac{\gamma_{0}}{\Gamma_{B}}\right)-\frac{1}{2} \frac{\gamma_{0}}{\gamma_{1}} \frac{1}{\Delta\left(F^{-}\right)}$
> >
> > where $\Delta(R) = R_0/R_1$ and $\Delta_l(R) = \log \Delta(R)$. The above point is accurate up to $O\left(\frac{\Gamma_B^2}{\gamma_1^2}\left(\frac{F^-_1}{F^-_0}\right)^2\right)$.
> >
> > ### Empirical Solution
> > We note that the threshold is a hyper-parameter. Assuming access to additional (possibly public) data $\mathcal{D}$, one can calibrate the threshold using a separate “calibration” dataset:
> >
> > 1. Train a state-of-the-art black-box model using $D_{train} \sim \mathcal{D}$
> > 2.Train an explainable model $M_{honest}$ on $D_{train} \sim \mathcal{D}$ without using any additional constraints on the gap between the black-box and interpretable model
> > 3. Train an explainable model $M_{fairwashed}$ on $D_{train} \sim \mathcal{D}$ using the Informed Adversary Optimization of Definition 5 in order to minimize the fairness gap
> > 4. Measure the KL divergence of $D_{sg} \sim \mathcal{D}$ on $M_{honest}$ and $M_{fairwashed}$ to form $X_{honest}$ and $X_{fairwashed}$ datasets. Assign labels $y = 1$ to $X_{fairwashed}$ and $y = 0$ to $x_{honest}$.
> > 5. $X= X_{honest} \cup X_{fairwashed}$, and $Y = Y_{honest} \cup Y_{fairwashed}$ form a univariate regression model with the following loss function $\ell$:
> > $\ell(x, y, T)=\sum_{i} \frac{1}{2}\mathbb{I}\left(x_i \leq T, y=1\right)+\frac{1}{2}\mathbb{I}\left(x_i>T, y=0\right)$. And the optimal threshold $T^* = \arg\min_T \ell(x, y, T)$.
> > Note that this assumes equal weights for type I and type II error of the detector. Finally, we note the state-of-the-art in calibration is [1]. However, a SOTA method is likely not necessary because if there are enough samples for calibration, central limit theorem ensures both  $X_{fairwashed}$ and $X_{honest}$ are normally distributed at which point the optimal threshold is simply $T^* = \frac{1}{2}(\mu_{fairwashed} +\mu_{fairwashed})$.
> >
> >
> > [1] R. Sahoo, A. Chen, S. Zhao, and S. Ermon, “Reliable Decisions with Threshold Calibration,” p. 14.

---

> > > ### Author Response · Authors · 2022-08-02
> > > **Response to Reviewer 53g2**
> > >
> > > > **Overall, I think this paper needs to much more rigorously provide the problem setting and explicit definitions of what is going on. Could you help clarify the problem setup a bit more, per the weaknesses section?**
> > >
> > > We clarified our assumptions, problem and solution based on the below points:
> > >
> > > Assumptions:
> > > We follow the standard assumptions made in the fairwashing literature:
> > > Company uses a black-box model $B$;
> > > Binary classification tasks, $B:\mathbb{R}^M \mapsto \{0,1\}$;
> > > Binary-valued sensitive attributes, $A \in \{0,1\}$.
> > >
> > >
> > > Problem statement:
> > > Below, we clarify the problem setup in 3 steps:
> > > The company trains $B$  on their training set $(X_{\text{Tr}}, Y_{\text{Tr}}, A)$, in which $A$ represents a sensitive attribute;
> > > The company makes decisions based on the prediction of $B$  about users' queries;
> > > After receiving decisions that they deem unfair, users from the suing set requires the company to be investigated in terms of the effect of $A$ on the decisions by an auditor;
> > > Given the black-box model $B(\cdot)$ and the suing set $X_{\text{sg}}, B(X_{\text{sg}},)$, company solves the fairwashing optimization (Definition 4) defined as learning an interpretable model $I(\cdot)$ from $B(\cdot)$ on $X_{sg}$ such that the interpretable model has 1) high fidelity with respect to the black-box model and 2) is less unfair than this black-box model.
> > >
> > >
> > > Our solution:
> > > To distinguish between the fairwashed and honest interpretable model in a non-cooperative way we propose to compute the divergence between subgroup confusion matrices $C_0$ and $C_1$ using Kullback–Leibler (KL) as $\mathcal{C}_{\text{KL}}=\text{KL}(C_0,C_1)$.
> > >
> > >
> > >
> > >
> > > > **Figure 3 is a bit hard to follow. Could you help clarify this figure? Where does the dotted line come from? Why are there multiple fidelity values for every Δ?**
> > >
> > > The dotted line is the unfairness of the black-box model computed on the suing set data. Figure 3 displays the results of solving the constrained optimization problem in Equation 9. More precisely, the constraints in Equation 9 are related to the fidelity (defined based on loss) and $\Delta$. For each value of $\Delta$, we consider different values for fidelity because our objective is to assess the evasion power of fairwashing attacks on the Rashomon Set of interpretable models. This is designed to characterize the damage an adversary can achieve given a constraint on $C_{KL}$. Therefore, the multiple fidelity values for every $\Delta$ in Figure 3 show the performance of the detector when facing different high-fidelity interpretable models.

---

> > > > ### Comment · Reviewer_53g2 · 2022-08-05
> > > > **Thanks for the comments**
> > > >
> > > > Thanks for getting back, and this helps clarify the paper considerably for me.
> > > >
> > > > I'm still about confused about the statement of Theorem 1. If it doesn't matter that I is an "interpretable model" why not just say a surrogate model trained to predict B's predictions? This is much more explicit than having terms like "interpretable" and "explain" in the theorem statement, which doesn't necessarily have concrete definitions, and you can describe later on the theorem applies in case I is a very simple interpretable model. I think using an argument like this would help clarity here.
> > > >
> > > > I think the paper is in better shape overall, however, so I upgraded my score to weak accept.

---

> > > > > ### Author Response · Authors · 2022-08-09
> > > > > **Thank you!**
> > > > >
> > > > > Thank you very much for your time reading our responses, upgrading your score and very insightful suggestions. Indeed, for the theorem to hold, the surrogate model does not need to be an interpretable model even if in practice it is almost always the case in the context of post-hoc explanation as the objective is precisely to explain a black-box. We will incorporate your suggestion in the final version of our manuscript by changing the first sentence of our theorem to
> > > > >
> > > > > ‘’Assume $\hat{Y}_{B}$ are the black-box model $B(\cdot)$ predictions, $\hat{Y}_I$ are the predictions of a surrogate (interpretable) model $I(\cdot)$ trained on  $B(\cdot)$ outputs and $A$ is a sensitive attribute:’’

---

### Official Review · Reviewer_bYAU · 2022-07-11

**Rating:** 3
**Confidence:** 4
**Soundness:** 3 good
**Presentation:** 3 good
**Contribution:** 1 poor

**Summary:**

The authors deal with the problem of detecting fairwashing, a maliciously generated explanation such that the model looks fair even if it is actually unfair. They investigated fairwashing in the global explanation and found that elimination of fairwashing is difficult, and fairwashing can be detected using false- and true- positive rates. Then, their proposed detection algorithm for fairwashing is to check the deviation of the false/true positive/negative rates between the original and explanation models using KL divergence. To evaluate the robustness of their algorithm, they introduce a fairwashing algorithm with a mechanism evading the proposed detection method. The empirical evaluations demonstrate the high detectability and robustness of their proposed detection algorithm.

**Questions:**

See strengths and weaknesses.

**Limitations:**

The accessibility to $B$ makes the situation no fairwashing risk. Hence, I recommend the authors reconsider the situation to be meaningful.

**Strengths And Weaknesses:**

(Strengths)
- The fairwashing is an urgent risk. Hence, the detection of the fairwashing is well motivated.
- To the best of my knowledge, this is the first to propose a detection method for fairwashing.

(Weakness)
- The authors make an impractical assumption about the defender's knowledge.
- Lacks some important references.

The authors deal with the interesting and demanding task of detecting fairwashing, and most of the results are technically sound. However, I found a fundamental flaw in the problem setting, by which they deal with the situation where there is no risk of fairwashing. Due to such a considerable flaw, I recommend the rejection of this paper.

The problem setting looks weird. The authors assume the detector can query the original model $B$ in a black-box way. However, if we can query $B$, we can also calculate the unfairness score of $B$, such as the difference of the conditional probabilities in the demographic parity, by which we can confirm the fairness of the original model $B$. Hence, under the assumption of having accessibility to $B$, the defender easily detects the unfairness of the original model $B$. In other words, there is no risk of fairwashing in this context. While this paper adequately validates the fact that their method can detect fairwashing in this context,  the detectability is obvious due to accessibility to $B$.

The statement of Theorem 1 is not rigorously defined. What mean by "completely eliminating fairwashing"? Also, what mean by "sufficient"?

This paper lacks the comparison with the following studies:
- K. Fukuchi, et al. "Faking fairness via stealthily biased sampling."  AAAI2020.
- D. Slack, et al. "Fooling lime and shap: Adversarial attacks on post hoc explanation methods." AIES2020.

Both papers investigate the risk of deceiving fairness. Remarkably, the first paper discusses the detectability of malicious modification for deceiving unfairness under situations where the detector can access the labeled benchmark dataset.

---

> ### Author Response · Authors · 2022-08-02
> **Response to Reviewer bYAU**
>
> We thank the reviewer for the comments and for highlighting that we are the first to propose a detection method for fairwashing.
>
>
> > **The problem setting looks weird. The authors assume the detector can query the original model B in a black-box way. However, if we can query B, we can also calculate the unfairness score of B, such as the difference of the conditional probabilities in the demographic parity, by which we can confirm the fairness of the original model B. Hence, under the assumption of having accessibility to B, the defender easily detects the unfairness of the original model
> B. In other words, there is no risk of fairwashing in this context. While this paper adequately validates the fact that their method can detect fairwashing in this context, the detectability is obvious due to accessibility to B.**
>
> Please note that the Algorithm 1 never accesses the black-box model after the first query step which is solely done on the suing set samples. We stress that this dataset is available before any formal audit takes place. This does _not_ constitute query access to the black-box model; in fact it is only dataset access, which here is the predictions on the suing set. Under this problem setting, there is a risk of fairwashing.
>
> We acknowledge a mistake in Algorithm 1’s input that mentioned query access (despite not using it on any sample other than the suing set samples). This has been fixed in the revised submission and is uploaded. We thank the reviewer for the insightful comment.
>
> > **The statement of Theorem 1 is not rigorously defined. What mean by "completely eliminating fairwashing"? Also, what mean by "sufficient"?**
>
> By “completely eliminating fairwashing” we mean to reduce the fairwashing gap $\gamma$ (defined in Equation 3) to zero. We discuss this phenomenon in the Irreducibility paragraph (line 202 and Equation 6) where we show that $\gamma = \Gamma_B (1 + F^+_1 - T^+_1) > 0$. We have adapted the revision text to highlight $\gamma > 0$.
>
> By sufficiency of measuring false-positive and true-positive rates of the interpretable model w.r.t black-box model for detecting fairwashing, we mean that only these two rates (plus the black-box model unfairness which is not a characteristic of the detector), fully characterize the fairwashing gap. That is, measuring any other metric (for instance, to create a new detector) is unnecessary. We show this in the Sufficiency paragraph (line 209).

---

> > ### Author Response · Authors · 2022-08-02
> > **Response to Reviewer bYAU**
> >
> > > **This paper lacks the comparison with the following studies:
> > [AAAI-2020] K. Fukuchi, et al. "Faking fairness via stealthily biased sampling." AAAI2020.
> > [AIES-2020] D. Slack, et al. "Fooling lime and shap: Adversarial attacks on post hoc explanation methods." AIES2020.
> > Both papers investigate the risk of deceiving fairness. Remarkably, the first paper discusses the detectability of malicious modification for deceiving unfairness under situations where the detector can access the labeled benchmark dataset.**
> >
> > Thank you for these suggestions. After verification, both of the suggested papers share indeed key similarities with our own in the sense that they consider deceptions of fairness. These papers also further highlight the importance of conducting further research in fairwashing, model auditing, as well as model explanation. We added the following discussion in the Related Work (Section 2.1) and Appendix B.
> >
> > To summarize, the first paper [AAAI-2020], similar to us, tries to detect fairwashing but with a different setting. The second paper [AIES-2020] shows an alternate method of fairwashing that evades model explanation techniques in a setting that is comparable to ours but does not attempt to detect the fairwashing. Next, we highlight some of the differences in settings, assumptions and solutions:
> >
> >
> > - Setting: We rely on explainer models while they use a published data subset for fairness auditing. In particular, they reveal some subset of the training data and their predictions. We follow the setting considered in the literature introducing fairwashing ([ICML-2019] and [NeurIPS-2021]). [AIES-2020] explanations require input perturbations while ours require the model owner to provide an interpretable model.
> >
> >
> > - Assumptions: [AIES-2020]  assume that fairness auditing is performed via model-agnostic local explanations (e.g., LIME and SHAP). Both [AIES-2020] and [AAAI-2020] assume an ideal detector with knowledge of the underlying training distribution of the model; and query access to the black-box model. However, we only assume access to black-box model predictions on the suing set. We stress that this dataset is available before any formal audit takes place (in the form of asking for model explanations). This does _not_ constitute query access to the black-box model; in fact it is only dataset access, which here is the predictions on the suing set.
> >
> >
> > - Solutions: They attempt to detect fairwashing using a Kolmogorov-Smirnov (KS) test over the underlying and company-provided subset distribution to determine if the data subset was honestly provided. More precisely, they show that detection is difficult when fairwashing is conducted by minimizing the Wasserstein distance between the distributions while subject to fairness; this optimization also minimizes the upper-bound of the advantage (i.e., distinguishability) of the KS test. This differs from our setting in which an informed malicious company must optimize the fairness subject to both fidelity (loss) and the detection threshold.
> >
> > Explanation methods like LIME and SHAP perturb inputs to gauge feature relevance, inadvertently querying with synthetic data that may be detected with out-of-distribution detection. Queries determined to originate from explainers are fed to a fair model whereas in-distribution points are given to the biased model. In contrast, our detection method is non-cooperative (we require no additional information from the black-box model as the auditor already has the predictions on the suing set) and therefore does not rely on input perturbations.

---

> > > ### Comment · Reviewer_bYAU · 2022-08-08
> > > **Thanks for your response**
> > >
> > > I thank the authors for your response.
> > >
> > > > Accessibility to $B$
> > >
> > > I'm still confident that the accessibility to $B$ is problematic. First, the theoretical analyses shown in Section 3 are dependent on the accessibility to the model $B$. As mentioned in the initial comments, since we can calculate the unfairness directly because of the accessibility to $B$, the detectability is obvious. Second, if we can obtain the suing dataset labeled by $B$, we can calculate the unfairness from the labeled dataset, meaning that we can easily get the unfairness of the original model. Hence, the detectability is trivial even in this case.
> > >
> > > > Sufficiency
> > >
> > > I understand the definition of "completely eliminating fairwashing." However, it is still unclear about the definition of sufficiency. In the authors' analysis, $\gamma$ involves terms other than the false- and true-positive rates. The authors claim in Line 209 that such other terms are constant. However, without a mathematically rigorous definition of sufficiency, I cannot confirm that these terms are actually independent of sufficiency.
> > >
> > > Due to these concerns, I'd like to keep my score unchanged.

---

> > > > ### Author Response · Authors · 2022-08-09
> > > > **Response to the remaining concerns (part 2 of 2)**
> > > >
> > > > > **Sufficiency
> > > > I understand the definition of "completely eliminating fairwashing." However, it is still unclear about the definition of sufficiency. In the authors' analysis, γ involves terms other than the false- and true-positive rates. The authors claim in Line 209 that such other terms are constant. However, without a mathematically rigorous definition of sufficiency, I cannot confirm that these terms are actually independent of sufficiency.**
> > > >
> > > > Regarding the definition of sufficiency, what we mean by sufficiency is simply that $\tilde{\delta}$ and $\delta’$ are sufficient for identification of fairwashing (i.e., an auditor does not require a greater amount of information than $\tilde{\delta}$ and $\delta’$ to determine whether fairwashing has occurred). We note that as the other components of the fairwashing gap $\gamma$ are irreducible ( $\Gamma_B (1 + F_1^{+} - T_1^{+})$ ), these cannot be helpful in the determination of fairwashing and thus are excluded from our sufficiency statement. As $\tilde{\delta}$ and $\delta’$ rely on the black-box and interpretable model while $\Pr\left[\hat{Y}_{B}=y \mid A=0\right], y\in\{0,1\}$ is exclusively related to the black-box model. We will clarify in the paper by stating that $\tilde{\delta}$ and $\delta’$ are sufficient information from the interpretable model to determine fairwashing, as the other terms are constant from the perspective of the interpretable model. Since we are attempting to determine fairwashing of the interpretable model, we discuss sufficiency w.r.t. that model. We intend sufficiency as in logical implication such that certain values of $\tilde{\delta}$ and $\delta’$ alone provide determination for fairwashing.
> > > >
> > > > Given the above, we propose the following definition of sufficiency:
> > > >
> > > > **Definition.** *We define sufficiency in the context of determination of fairwashing as the dependence of fairwashing on particular variables – i.e. the values taken by particular variables form a sufficient condition for the determination of fairwashing. In our case, if the values of $\tilde{\delta}$ and $\delta’$ exceed a threshold, this is a sufficient condition for fairwashing.*

---

> > > > ### Author Response · Authors · 2022-08-09
> > > > **Response to the remaining concerns (part 1 of 2)**
> > > >
> > > > Thank you for your time reading our responses and posting new comments. We respond to them inline.
> > > >
> > > > > **Accessibility to B.
> > > > I'm still confident that the accessibility to B is problematic. First, the theoretical analyses shown in Section 3 are dependent on the accessibility to the model B. As mentioned in the initial comments, since we can calculate the unfairness directly because of the accessibility to B, the detectability is obvious. Second, if we can obtain the suing dataset labeled by B, we can calculate the unfairness from the labeled dataset, meaning that we can easily get the unfairness of the original model. Hence, the detectability is trivial even in this case.**
> > > >
> > > > Regarding the accessibility to B, we would like to highlight that we did not invented a new problem setting, rather we have followed the problem setting that was previously introduced in the seminal papers of the fairwashing literature. See for instance :
> > > >
> > > > - [ICML-2019] Ulrich Aïvodji, Hiromi Arai, Olivier Fortineau, Sébastien Gambs, Satoshi Hara, Alain Tapp: Fairwashing: the risk of rationalization. ICML 2019: 161-170.
> > > >
> > > > - [NeurIPS-2021] Ulrich Aïvodji, Hiromi Arai, Sébastien Gambs, Satoshi Hara: Characterizing the risk of fairwashing. NeurIPS 2021: 14822-14834.
> > > >
> > > > We agree with the reviewer that we can compute the unfairness of the black-box model and interpretable model on the suing set. However, these unfairness values (or their differences) cannot be used to detect whether the interpretable model is adversarially manipulated to rationalize decisions taken by the black-box (i.e. fairwashing). This is due to the fact that the unfairness value alone provides little insight on which design choice (e.g., a fairness constraint was added to the optimization objective) introduced the bias (with respect to the sensitive attribute). In addition to this, we show both empirically and theoretically that the unfairness of the black-box model and the unfairness of its surrogate model are always different. We mentioned this in lines 58-61 of our introduction as:
> > > >
> > > > *“measuring the differences between the fairness of the black-box model and the explainable model cannot help to detect fairwashing as there are several design choices (including fairwashing) that could lead to different fairness, thus non-intentional fairwashing.”*
> > > >
> > > > Hereafter, we illustrate why this is the case with an example.
> > > >
> > > > 1. A company uses a black-box model to provide personalized services or make decisions on users queries, for example deciding based on the application of a user if they can get a loan. Here, the black-box model’s prediction would reflect the company’s decision in response to the loan application.
> > > >
> > > > 2. Upon receiving the company’s decision, some users may deem that the decision they received is biased. For example, they may believe that it is unfair with respect to the “gender” sensitive attribute. As a consequence, users report this issue to an auditor by sending their data along with the decision they received. As multiple user complaints accumulate, a suing set is built (e.g., for a class action).
> > > >
> > > > 3. We agree with the reviewer that the auditor can compute the black-box model’s unfairness on the suing set. However, the unfairness value alone provides little insight on which design choice introduced the bias (with respect to the sensitive attribute). For instance, in the scenario considered, evaluating the unfairness of predictions made on the suing set alone does not characterize the impact of “gender” on the predictions of the black-box model. To achieve this, the Explainable AI community has developed post-hoc explanation techniques as a way to assess the fairness of models by explaining the “reasoning” behind the predictions of the model as well as its reliance on sensitive attributes. Therefore, in the problem setup proposed by previous papers on fairwashing, the auditor asks for a model explanation from the company to determine whether the “gender” attribute has a direct impact on the model predictions. As our detector is non-cooperative so it does not need further query access to the black-box model than what was already collected by users on the suing set (put another way, our detector does not need to perform any additional adaptive queries to the black-box beyond the initial suing set).
> > > >
> > > > 4. In order to evade penalties from the auditor, the company trains a fairwashed interpretable model in which the impact of “gender” on the model prediction is fairwashed (i.e., decreased) and sends this model to the auditor.
> > > >
> > > > 5. If the company successfully fairwashed their model, upon inspecting the interpretable model, the auditor gets a false impression that “gender” is not impacting the prediction. Instead, the auditor finds that other attributes such as “occupation” strongly affect the output. The company has successfully evaded accusations of unfairness with respect to the “gender” attribute.
> > > >
> > > > This is precisely what we detect with our work.

---

### Official Review · Reviewer_vBkK · 2022-07-14

**Rating:** 6
**Confidence:** 4
**Soundness:** 4 excellent
**Presentation:** 3 good
**Contribution:** 4 excellent

**Summary:**

This paper focuses on the problem of fairwashing detection in black-box models. To be specific, fairwashing is a technique used to fool model explanation methods to produce deceptive outputs for unfair models. In this paper, the authors argue the importance of studying the faiwashing issue, and propose a novel framework FRAUD-Detect to detect fairwashed models, and present empirical results. In addition, theoretical analysis is provided to show that fairwashing is unlikely to be avoided. And the robustness of the proposed detection method towards an informed adversary is also discussed.

**Questions:**

1. How should we choose the hyperparameter \delta in different scenarios (e.g. different tasks, different data)?

**Limitations:**

No potential negative societal impact.

**Strengths And Weaknesses:**

Strengths:
1. This paper investigates a novel and important problem. Model explanation is a popular and promising approach for auditing a model with critical usage to avoid unfairness in automated decision-making. But such model explanation approaches are also vulnerable to being attacked, which has been overlooked. The problem is also challenging, a thorough study on it is needed.

2. The paper provides a simple but effective method for detecting dishonest interpretation models that take advantage of the fairwashing technique.

3. Theoretical conclusion is provided to show that fairwashing is unlikely to be completely avoided. Empirical results are provided to validate the effectiveness of the proposed detection method and also its robustness towards an informed attack.

Weaknesses:
1. The logic of the paper is clear but the expressions in the paper are kind of obscure and not easy for readers to follow. The motivation and the algorithm in the proposed detection model are simple and clear, but the authors use a lot of long sentences to describe them, which makes the simple things complicated. It would be better if the authors could use more short and succinct sentences, which will improve the presentation and readability of the paper.

2. The hyperparameter \delta (the fairwashing detection threshold) is a key parameter in this proposed detection algorithm. More discussions on how to choose this hyperparameter in practical use are needed.

---

> ### Author Response · Authors · 2022-08-02
> **Response to Reviewer vBkK**
>
> We thank the reviewer for the comments and for highlighting the importance of our analytical and empirical analysis.
>
> > **The logic of the paper is clear but the expressions in the paper are kind of obscure and not easy for readers to follow. The motivation and the algorithm in the proposed detection model are simple and clear, but the authors use a lot of long sentences to describe them, which makes the simple things complicated. It would be better if the authors could use more short and succinct sentences, which will improve the presentation and readability of the paper.**
>
> Thanks for this suggestion. We have implemented it in the revised version of the paper, in particular by improving the presentation and splitting long sentences into short and succinct sentences. The revised version of the manuscript has been uploaded.
>
>
> > **The hyperparameter \delta (the fairwashing detection threshold) is a key parameter in this proposed detection algorithm. More discussions on how to choose this hyperparameter in practical use are needed.
> How should we choose the hyperparameter \delta in different scenarios (e.g. different tasks, different data)?**
>
> We thank the reviewer for the insightful comment. The question of choosing a threshold is a pertinent one. We provide two answers, one based on the theory we have presented in Section 3, and an empirical solution.
>
> ### Theory Solution
> In Appendix I, we provide the following bounds for the KL divergence measure of FRAUD-detect:
>
> $\kappa := \operatorname{KL}\left(
>         \begin{bmatrix} F^+_0 \\
>                         F^-_0
>         \end{bmatrix},
>         \begin{bmatrix} F^+_1 \\
>                         F^-_1
>         \end{bmatrix}
>         \right)$
>
> is bounded from below and above by:
>
> $
>   F_0^+ \log\left(\frac{F_0^+}{F_1^-} \cdot \frac{\gamma_0}{\gamma_1}\right)
>                         \leq  \kappa \leq
>   F_0^- \log\left(\frac{\gamma_0}{\gamma_1} \cdot \frac{F_0^-}{F_1^+}\right)
> $
>
> Any choice between
> $\kappa_{min} := F_0^+ \log\left(\frac{F_0^+}{F_1^-} \cdot \frac{\gamma_0}{\gamma_1}\right)$
> and
> $\kappa_{max} := F_0^- \log\left(\frac{\gamma_0}{\gamma_1} \cdot \frac{F_0^-}{F_1^+}\right)$ is valid.
> $\kappa = \kappa_{min}$ is the most conservative choice which increases the risk of false discovery (of fairwashing), while $\kappa = \kappa_{max}$ is a permissive choice that minimizes the risk of false discovery at the expense of increase in the false negatives (not detecting fairwashing). A balanced choice for threshold  could be the intermediate point:
>
> $\bar{\kappa}=\frac{F_{0}}{2}\left(\Delta_{l}\left(F^{+}\right)+\Delta_{l}\left(F^{-}\right)\right)+\frac{\gamma_{0}}{2 \Gamma_{B}} \log \left(\frac{\gamma_{0}}{\Gamma_{B}}\right)-\frac{1}{2} \frac{\gamma_{0}}{\gamma_{1}} \frac{1}{\Delta\left(F^{-}\right)}$
>
> where $\Delta(R) = R_0/R_1$ and $\Delta_l(R) = \log \Delta(R)$. The above point is accurate up to $O\left(\frac{\Gamma_B^2}{\gamma_1^2}\left(\frac{F^-_1}{F^-_0}\right)^2\right)$.
>
> ### Empirical Solution
> We note that the threshold is a hyper-parameter. Assuming access to additional (possibly public) data $\mathcal{D}$, one can calibrate the threshold using a separate “calibration” dataset:
>
> 1. Train a state-of-the-art black-box model using $D_{train} \sim \mathcal{D}$
> 2.Train an explainable model $M_{honest}$ on $D_{train} \sim \mathcal{D}$ without using any additional constraints on the gap between the black-box and interpretable model
> 3. Train an explainable model $M_{fairwashed}$ on $D_{train} \sim \mathcal{D}$ using the Informed Adversary Optimization of Definition 5 in order to minimize the fairness gap
> 4. Measure the KL divergence of $D_{sg} \sim \mathcal{D}$ on $M_{honest}$ and $M_{fairwashed}$ to form $X_{honest}$ and $X_{fairwashed}$ datasets. Assign labels $y = 1$ to $X_{fairwashed}$ and $y = 0$ to $x_{honest}$.
> 5. $X= X_{honest} \cup X_{fairwashed}$, and $Y = Y_{honest} \cup Y_{fairwashed}$ form a univariate regression model with the following loss function $\ell$:
> $\ell(x, y, T)=\sum_{i} \frac{1}{2}\mathbb{I}\left(x_i \leq T, y=1\right)+\frac{1}{2}\mathbb{I}\left(x_i>T, y=0\right)$. And the optimal threshold $T^* = \arg\min_T \ell(x, y, T)$.
> Note that this assumes equal weights for type I and type II error of the detector. Finally, we note the state-of-the-art in calibration is [1]. However, a SOTA method is likely not necessary because if there are enough samples for calibration, central limit theorem ensures both  $X_{fairwashed}$ and $X_{honest}$ are normally distributed at which point the optimal threshold is simply $T^* = \frac{1}{2}(\mu_{fairwashed} +\mu_{fairwashed})$.
>
>
> [1] R. Sahoo, A. Chen, S. Zhao, and S. Ermon, “Reliable Decisions with Threshold Calibration,” p. 14.

---

> > ### Author Response · Authors · 2022-08-09
> > **Thank you!**
> >
> > Dear reviewer vBkK,
> >
> > Thank you very much for the very positive feedback and insightful suggestions! We would be happy to answer any further questions you may have before the response period ends today.
> >
> > Warm regards,
> > Paper3526 Authors

---

### Author Response · Authors · 2022-08-05
**Thank you! we are available for discussion**

Dear reviewers,

Thank you very much for your time serving as the reviewer for our paper! We would be happy to answer any further questions you may have before the response period ends on Aug 9.



Warm regards,

Paper3526 Authors

---

### Comment · Area_Chair_HyHc · 2022-08-07
**Discussion with Authors**

Dear Reviewers! Thank you so much for your time on this paper so far.

The authors have written a detailed response to your concerns. How does this change your review?

Please engage with the authors in the way that you would like reviewers to engage your submitted papers: critically and open to changing your mind. Thank you Reviewers 53g2 and WPPT for your initial engagement!

Looking forward to the discussion!

---

### Meta-Review · Area_Chair_HyHc · 2022-08-26

**Recommendation:** Accept
**Confidence:** Less certain

**Metareview:**

The reviewers were split about this paper: on one hand they appreciated the motivation and the comprehensive experiments in the paper, on the other they were concerned about the clarity of the paper, even worried about a potential flaw. I have decided to vote to accept given the clear and convincing author response. I urge the authors to take all of the reviewers changes into account (if not already done so). Once done this paper will be a nice addition to the conference!

**Award:**

No

---

### Decision · Program_Chairs · 2022-09-14

Accept